# Personalized Subgraph Federated Learning with Differentiable Auxiliary Projections

**Wei Zhuo**[1], **Zhaohuan Zhan**[2], **Han Yu**[1]

[1]Nanyang Technological University, [2]Shenzhen MSU-BIT University

[1]{wei.zhuo, han.yu}@ntu.edu.sg, [2]zhan.z@smbu.edu.cn

## Abstract

Federated Learning (FL) on graph-structured data typically faces non-IID challenges, particularly in scenarios where each client holds a distinct subgraph sampled from a global graph. In this paper, we introduce **Fed**erated learning with **Aux**iliary projections (`FedAux`), a personalized subgraph FL framework that learns to align, compare, and aggregate heterogeneously distributed local models without sharing raw data or node embeddings. In `FedAux`, each client jointly trains (i) a local GNN and (ii) a learnable auxiliary projection vector (APV) that differentiably projects node embeddings onto a 1D space. A soft-sorting operation followed by a lightweight 1D convolution refines these embeddings in the ordered space, enabling the APV to effectively capture client-specific information. After local training, these APVs serve as compact signatures that the server uses to compute inter-client similarities and perform similarity-weighted parameter mixing, yielding personalized models while preserving cross-client knowledge transfer. Moreover, we provide rigorous theoretical analysis to establish the convergence and rationality of our design. Empirical evaluations across diverse graph benchmarks demonstrate that `FedAux` substantially outperforms existing baselines in both accuracy and personalization performance. The code is available at https://github.com/JhuoW/FedAux.

## 1 Introduction

Real-world data often manifests as relational structures, ranging from social interactions [30, 46] and financial networks [28, 45] to molecular graphs [35, 44], whose scale and privacy constraints increasingly require training to be carried out in a federated manner [9], whereby multiple clients collaboratively learn a Graph Neural Network (GNN) model without exchanging their raw data. However, applying federated learning to graph-structured data, such as social networks, faces severe challenges due to *non-identically and independently distributed* (non-IID) data across clients. For example, consider a federated learning scenario involving multiple regional social networking platforms, each representing a distinct subgraph of a global social network. Users within each region exhibit unique interaction patterns and distinct interests, resulting in significant heterogeneity in local graph structures and node attributes. This inherent diversity among subgraphs leads to substantial difficulties when attempting to aggregate local GNN models into a unified global model, as traditional FL algorithms [20, 14] typically assume homogeneous data distributions across clients.

To tackle the non-IID challenges inherent in subgraph federated learning, personalized FL [29] has recently emerged as a promising paradigm, which aims to provide client-specific GNN models rather than enforcing a universal global solution. Existing personalized subgraph FL approaches commonly achieve personalization by clustering clients on the server side, necessitating a reliable measure of client similarity without direct access to client-side data. In this work, we impose even stricter privacy constraints: neither raw data nor embeddings are shared, and only model learnable parameters can be exchanged. Although the server could compare clients by directly measuring similarity between their

39th Conference on Neural Information Processing Systems (NeurIPS 2025).

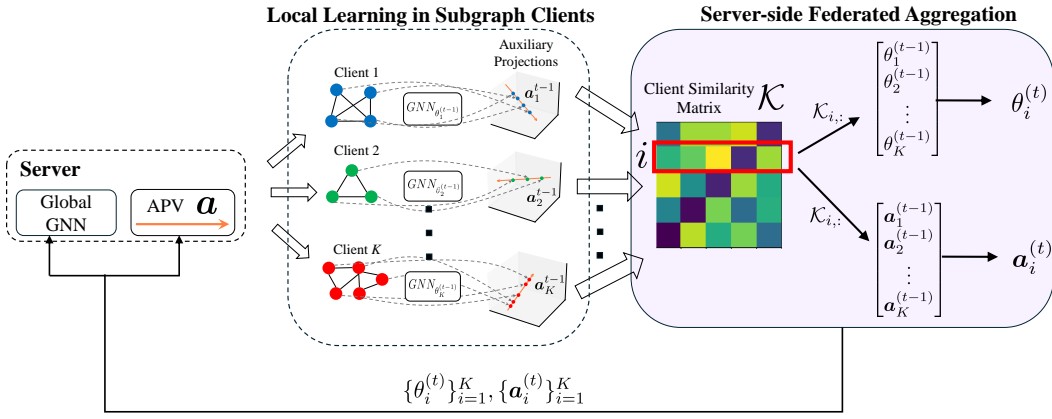

**Local Learning in Subgraph Clients**

**Server-side Federated Aggregation**

**Update Global Parameters**

Figure 1: The overall framework of `FedAux`. Left: The server maintains a global GNN model together with learnable auxiliary projection vectors (APVs) that are broadcast to all clients at the start of each communication round. Middle: Clients jointly optimize the GNN and APV during local training, where the APV projects node embeddings onto a 1D $a_k$-space that positions related nodes closer together. Right: After local training, clients transmit their optimized GNN parameters and personalized APVs to the server. The server computes a client similarity matrix by comparing the learned APVs, which captures the heterogeneity across subgraphs without accessing raw data. These similarities determine personalized aggregation weights. At the end of each round, the updated global parameters are broadcast for the next communication round.

parameter matrices uploaded, the high dimensionality of these matrices makes such metrics unreliable under the curse of dimensionality [4]. Recent improvements have proposed measuring similarity by comparing communication-level parameter gradients [35] or generating a common anchor graph on the server as a neutral testbed [3]. Although these strategies mitigate some limitations, they remain largely heuristic and do not explicitly model the heterogeneity inherent in subgraph clients (See extended discussion in Appendix A).

**Motivation** Our key insight is that a compact, low-dimensional proxy, derived directly from the client's own model parameters, can faithfully summarize local subgraph characteristics without leaking sensitive node features or embeddings. Such a proxy remains compact enough to avoid the pitfalls of high-dimensional similarity measures, yet expressive enough to reflect meaningful differences between clients. By learning this proxy jointly with the GNN parameters in each client, and using it to guide both local adaptation and server-side aggregation, we obtain a principled, privacy-preserving mechanism for personalization that directly leverages model parameters as a stand-in for subgraph information.

**Contribution** In this work, we propose `FedAux`, which employs differentiable auxiliary projections to effectively capture and exploit client-specific heterogeneity for subgraph FL. As illustrated on the left of Fig. 1, the server stores not only a global GNN but also a learnable auxiliary projection vector (APV) that accompanies the model parameters. At the start of the first communication round, the server broadcasts the global GNN and the current APV to all clients. Each client projects its node embeddings onto the APV, which is treated as a one-dimensional latent space. A differentiable soft-sorting operator then orders the projected embeddings by similarity, after which a simulated 1D convolution aggregates the sorted embedding sequence. The aggregated representations drive a supervised loss that simultaneously refines the local GNN and the APV, so that the optimized APV preserves the relational structure of the client subgraph. Upon completing local training, clients send their updated GNN weights and personalized APVs back to the server. Since the APV reveals only the latent space that best preserves local node relationships while concealing the exact position of every node in this space, it acts as a compact privacy-preserving summary of the client subgraph. Then the server computes similarity among the returned APVs to quantify inter-client affinity and yields client-specific aggregation weights. The server uses these weights to combine the incoming parameters, producing a personalized model for each client that respects both shared knowledge and local subgraph idiosyncrasies.

Furthermore, we establish comprehensive and rigorous theoretical analyses that justify the soundness and interoperability of every technique used in `FedAux`. Extensive federated node classification experiments on six datasets, spanning diverse graph domains and client scales, demonstrate that `FedAux` achieves better accuracy and stronger personalization than state-of-the-art personalized subgraph FL baselines.

## 2  Problem Statement: Subgraph Federated Learning

In Federated Learning (FL), multiple clients collaboratively train a global model without exchanging their raw data. In the *subgraph federated learning* setting, each client holds a subgraph of a larger graph. Formally, let a graph $\mathcal{G}$ be partitioned (or subsampled) into $K$ subgraphs $\{G_1, G_2, \cdots, G_K\}$ as $K$ clients, where $G_k = (V_k, E_k, X_k, Y_k)$. Here, $V_k = \{v_{k,1}, \cdots, v_{k,N_k}\}$ is the set of nodes in the $k$-th subgraph with $N_k = |V_k|$ nodes, $E_k$ the set of edges among those nodes, $X_k \in \mathbb{R}^{N_k \times d}$ the node features, and $Y_k$ the labels relevant to the learning task. In our FL scenario, each client $G_i$ has access only to its local data (i.e., its subgraph structure, node features, and labels), and there is no sharing of raw data or any node embeddings between clients.

A typical GNN $f_{\theta_k}(G_k)$ parameterized by $\theta_k$ is employed to produce node embeddings and ultimately generate predictions on the client $G_k$. In a standard federated learning setting such as FedAvg [20], one aims to solve the global objective: $\min_\theta \sum_{k=1}^K \alpha_k \mathcal{L}_k(\theta)$, subject to the privacy constraint that raw local data $G_k$ never leaves the client side. A common choice is to weight client $G_k$ by $\alpha_k = N_k / \left( \sum_{j=1}^K N_j \right)$ or simply $\alpha_k = 1/K$. The iterative procedure proceeds as follows. First, the server initializes $\theta^{(0)}$. At each global communication round $t \in \{1, \cdots, T\}$, it sends $\theta^{(t-1)}$ to each client. $G_k$ then updates $\theta^{(t-1)}$ locally by taking a few stochastic gradient steps on $\mathcal{L}_k(\theta)$ to update the parameters $\theta_k \leftarrow \theta_k - \eta \nabla \mathcal{L}$, which produce $G_k$'s optimal local parameters $\theta_k^{(t)}$. After the $t$-th local training, all clients' locally updated parameters $\{\theta_1^{(t)}, \cdots \theta_K^{(t)}\}$ are sent back to the server, which aggregates them via a weighted average $\theta^{(t)} = \sum_{k=1}^K \alpha_k \theta_k^{(t)}$. The newly aggregated global parameters $\theta^{(t)}$ are then broadcast back to each client for the next communication round. When the process converges or reaches a designated number of rounds, the final global parameters $\theta^{(T)}$ are taken as the parameters of the GNN model on the server.

## 3  Methodology

In this section, we introduce the proposed Subgraph Federated Learning with Auxiliary Projections (`FedAux`) framework, designed to address the heterogeneity across local subgraphs in federated learning. Fig. 1 illustrates an overview of `FedAux`. Our objective is twofold: 1) Each client locally encodes its subgraph into a one-dimensional space via a learnable auxiliary projection vector `APV`, and 2) the server then exploits these auxiliary vectors to realize personalized aggregation.

### 3.1  Client-Side Local Training

Before the first communication round $t = 0$, alongside the GNN model parameterized by $\theta^{(0)}$, the server also maintains a learnable `APV` $\boldsymbol{a}^{(0)} \in \mathbb{R}^{d'}$. During each communication round $t$, the server distributes $(\theta^{(t-1)}, \boldsymbol{a}^{(t-1)})$ to initialize all clients' local model $\{(\theta_k^{(t-1)}, \boldsymbol{a}_k^{(t-1)})\}_{k=1}^K \leftarrow (\theta^{(t-1)}, \boldsymbol{a}^{(t-1)})$. For a client $G_k$, it runs the local GNN model to optimize the node embeddings:

$$\mathbf{H}_k^{(t-1)} = f_{\theta_k^{(t-1)}}(G_k) = \left[ h_{k,1}^{(t-1)}, h_{k,2}^{(t-1)}, \cdots, h_{k,N_k}^{(t-1)} \right] \in \mathbb{R}^{N_k \times d'}, \tag{1}$$

where $d'$ is the output dimension of node embeddings. Given the local `APV` $\boldsymbol{a}_k^{(t-1)}$, we first normalize all node embeddings so that all embeddings are compared on a consistent scale as $\hat{h}_{k,i}^{(t-1)} = h_{k,i}^{(t-1)} / \max_j \|h_{k,j}^{(t-1)}\|$. Then the similarity between node $v_{k,i}$ and $\boldsymbol{a}_k^{(t-1)}$ is defined as $s_{k,i}^{(t-1)} = \langle \hat{h}_{k,i}^{(t-1)}, \boldsymbol{a}_k^{(t-1)} \rangle$, where $\langle \cdot, \cdot \rangle$ denotes the inner product in $\mathbb{R}^{d'}$. Intuitively, $s_{k,i}^{(t-1)}$ can be interpreted as the coordinate of each node $v_{k,i}$ in $\boldsymbol{a}_k$-space, which is a 1D line. Since $\boldsymbol{a}_k^{(t-1)}$ is itself learnable, the client $G_k$ is adaptively refining this space to capture relationships among its node embeddings more effectively.

Next, $G_k$ collects the similarity scores $S_k^{(t-1)} = \{s_{k,1}^{(t-1)}, \cdots, s_{k,N_k}^{(t-1)}\}$ and sort them in non-decreasing order. Let $\pi_k$ be the permutation that orders these scores:

$$s_{k,\pi_k(1)}^{(t-1)} \leq s_{k,\pi_k(2)}^{(t-1)} \leq \cdots \leq s_{k,\pi_k(N_k)}^{(t-1)}, \tag{2}$$

where $\pi_k(j)$ represents the node index at rank $j$. Accordingly, we apply this permutation $\pi_k$ to the row indices of $\mathbf{H}_k^{(t-1)}$, which yields the sorted embedding matrix $a_k^{(t-1)}$ as:

$$\widetilde{\mathbf{H}}_k^{(t-1)} = \left[\mathbf{H}_k^{(t-1)}\right]_{\pi_k,:} = \left[h_{k,\pi_k(1)}^{(t-1)}, h_{k,\pi_k(2)}^{(t-1)}, \cdots, h_{k,\pi_k(N_k)}^{(t-1)}\right], \tag{3}$$

which aligns node embeddings according to their coordinates in the $a_k$-space. As shown in Fig. 2, during local training, the node embeddings and the structure of the $a_k$-space are jointly optimized so that nodes with stronger relationships are positioned closer along this learned space (in $G_k$ comprising two triangles, nodes within the same triangle should be proximate in the $a_k$-space). In other words, the objective of the local model is to adaptively reshape the APV so that the induced node sorting effectively captures the local data information.

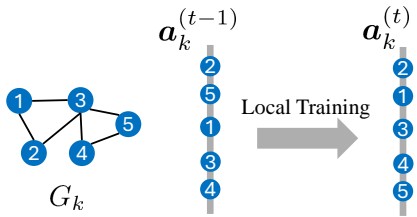

Figure 2: The local training of FedAux aims to map all nodes in $G_k$ to a corresponding $a_k$-space, and the optimization objective is to learn the APV $a_k$, such that the resulting $a_k$-space preserves the optimal node sorting.

Under the semi-supervised setting, the node sorting on APV can be adaptively refined under the guidance of the downstream task. Thinking of each $h_{k,\pi_k(i)}^{(t-1)}$ as a feature vector in a 1D sequence $\widetilde{\mathbf{H}}_k^{(t-1)}$, inspired by [18], we can apply a 1D convolution with a fixed kernel size $B$ over $\widetilde{\mathbf{H}}_k^{(t-1)}$ as $\mathrm{Conv1D}\left(\left[h_{k,\pi_k(1)}^{(t-1)}, h_{k,\pi_k(2)}^{(t-1)}, \cdots, h_{k,\pi_k(N_k)}^{(t-1)}\right]\right)$. More specifically, for each node $v_{\pi_k(i)}$ in the sorted sequence, the convolution can be written as:

$$z_{k,\pi_k(i)}^{(t-1)} = \sum_{\tau=-\lfloor B/2 \rfloor}^{\lfloor B/2 \rfloor} \mathbf{W}_\tau h_{k,\pi_k(i+\tau)}^{(t-1)} + b, \tag{4}$$

where $\mathbf{W}_\tau \in \mathbb{R}^{d' \times d'}$ are learnable convolution kernels for offset $\tau$, and $b$ is a bias term. Convolving around $v_{\pi_k(i)}$ in Eq. (4) amounts to a proximity-based aggregation in the $a_k$-space, where each embedding is updated by aggregating information from its neighbors along this learned 1D sorting. Hence, the quality of this sorting significantly impacts the aggregation effectiveness. Intuitively, nodes with stronger semantic relationships or similar labels should appear closer together in this learned sorting. We can formulate the learning objective to explicitly optimize the sorting induced by the APV $a_k$, ensuring that the resultant sorting facilitates effective aggregation and improves downstream predictive accuracy. Consequently, we formulate the learning objective:

$$(\theta_k^*, a_k^*, \Phi_k^*) = \underset{\theta_k, a_k, \Phi_k}{\arg\min} \mathcal{L}\left(\mathrm{CLF}\left(\mathrm{Conv1D}\left(\widetilde{\mathbf{H}}_k^{(t-1)}\right)\right), Y_k\right), \tag{5}$$

where $\Phi_k$ denotes the full set of parameters for $\mathrm{Conv1D}$ and the subsequent classifier $\mathrm{CLF}$ that maps node embeddings to final logits. Through this objective, we explicitly encourage $a_k$-space to yield an optimal node sorting, enabling the convolutional operation to effectively capture and leverage localized, label-informed relationships, thereby the optimized APV $a_k^{(t)} = a_k^*$ accurately preserves and encodes the local node relationships specific to each client.

However, $a_k$ does not directly participate in the loss defined in Eq. (5) in a way that enables standard backpropagation to update it. It is because the role of $a_k$ is limited to generating similarity scores, which in turn determine the input order to the $\mathrm{Conv1D}$ layer. Thus, $a_k$ only affects the network's output by reordering embeddings, which is a purely indirect pathway that does not produce a gradient signal for $a_k$ from the downstream loss. [18] attempted to mitigate this by multiplying each node embedding by its similarity score and then sorting. While this modification integrates $a_k$ into the learning pipeline directly, the hard discrete sort persists, causing the gradient signal that could refine $a_k$ to be still routed through a non-smooth transformation. Hence $a_k$ still cannot be fully optimized to reorder embeddings based on loss feedback, leaving the core issue unresolved.

To eliminate the hard-sorting bottleneck, we propose a *continuous aggregation* scheme over the $\boldsymbol{a}_k$-space. Rather than ranking or discretizing these similarity scores, for each node $v_i$, we define a continuous kernel $\kappa(s_{k,i}^{(t-1)}, s_{k,j}^{(t-1)})$, which could be a simple Gaussian-like function $\mathcal{K}_{ij} = \kappa(s_{k,i}^{(t-1)}, s_{k,j}^{(t-1)}) = \exp\left(-(s_{k,i}^{(t-1)} - s_{k,j}^{(t-1)})^2/\sigma^2\right)$ with bandwidth $\sigma > 0$, measuring how close $v_j$ is to $v_i$ in the real line spanned by $\{s_k^{(t-1)}\}$. As shown in the right part of Fig. 1, we then obtain an aggregated embedding for each node $v_i$ by a smooth weighted sum of all node embeddings:

$$z_{k,i}^{(t-1)} = \frac{1}{M_i} \sum_{j=1}^{N_k} \kappa\left(s_{k,i}^{(t-1)}, s_{k,j}^{(t-1)}\right) h_{k,j}^{(t-1)}, \quad M_i = \sum_{j=1}^{N_k} \kappa\left(s_{k,i}^{(t-1)}, s_{k,j}^{(t-1)}\right). \quad (6)$$

Unlike discrete sorting, this continuous aggregator is fully differentiable with respect to $\boldsymbol{a}_k$, because changes in $\boldsymbol{a}_k$ smoothly shift each $s_{k,i}^{(t-1)}$ and thus adjust the kernel weights $\kappa$. This approach naturally learns to group nodes with similar $s_k^{(t-1)}$ values, emulating the sorted 1D convolution effect, while sidestepping the gradient-blocking issues that arise from hard-sorting steps.

To jointly train the GNN parameters $\theta_k$ and the APV $\boldsymbol{a}_k$, we associate each node $v_i$ with two embeddings: $h_{k,i}^{(t-1)}$ produced by the GNN defined in Eq. (1), and $z_{k,i}^{(t-1)}$ generated via our kernel-based aggregation method as Eq. (6). We then concatenate these embeddings to form $v_i$'s final embedding $r_{k,i}^{(t-1)} = [h_{k,i}^{(t-1)} \| z_{k,i}^{(t-1)}]$, which is fed into a simple MLP classifier $\mathrm{CLF}(\cdot)$ to produce logits for the cross-entropy loss $\mathcal{L}_k = \frac{1}{N_k} CE(\mathrm{CLF}(\Gamma_k^{(t-1)}), Y_k)$, where $\Gamma^{(t-1)} = [r_{k,i}^{(t-1)}]_{i=1}^{N_k}$.

### 3.2 Server-Side Federated Aggregation

At the end of local training for communication round $t$, each client $G_k$ transmits its optimized parameters $(\theta_k^{(t-1)}, \boldsymbol{a}_k^{(t-1)})$ to the server. In doing so, only these high-level parameters are exchanged, rather than gradients or node embeddings, thereby limiting direct leakage of private subgraph information. Note that $\boldsymbol{a}_k$ **serves as the optimal subspace for capturing node relationships, how individual nodes map into this space (and thus the precise relational details) remains unknown to the server**. This design strictly adheres to the fundamental FL principle that *Data stays local; only model updates leave*.

Since the data distributions across clients can be non-IID, the server is expected to personalize the aggregation for each client. Given that each $\boldsymbol{a}_k^{(t-1)}$ can be a descriptor of how node embeddings in $G_k$ are arranged and structured, the similarity between two clients $G_k$ and $G_l$ can be measured via the cosine similarity of their APVs: $\mathrm{SIM}(\boldsymbol{a}_k^{(t-1)}, \boldsymbol{a}_l^{(t-1)}) = \frac{\langle \boldsymbol{a}_k^{(t-1)}, \boldsymbol{a}_l^{(t-1)} \rangle}{\|\boldsymbol{a}_k^{(t-1)}\| \|\boldsymbol{a}_l^{(t-1)}\|}$. We then convert this similarity into a weight:

$$w_{k,l}^{(t-1)} = \frac{\exp\left(\alpha \mathrm{SIM}\left(\boldsymbol{a}_k^{(t-1)}, \boldsymbol{a}_l^{(t-1)}\right)\right)}{\sum_{r=1}^{K} \exp\left(\alpha \mathrm{SIM}\left(\boldsymbol{a}_k^{(t-1)}, \boldsymbol{a}_r^{(t-1)}\right)\right)}, \quad (7)$$

where $\alpha > 0$ is a temperature controlling the sharpness of the weighting distribution. $w_{k,l}^{(t-1)}$ reflects how much client $G_k$ should incorporate the update from $G_l$. By emphasizing contributions from similar clients (i.e., those with high similarity in their APVs), each client's final model can better handle heterogeneous data while reducing interference from dissimilar clients. Instead of averaging all local parameters into one single global model, the server can compute a personalized aggregation of parameters for each client $G_k$ as:

$$\theta_k^{(t)} = \sum_{l=1}^{K} w_{k,l}^{(t-1)} \theta_l^{(t-1)}, \quad \boldsymbol{a}_k^{(t)} = \sum_{l=1}^{K} w_{k,l}^{(t-1)} \boldsymbol{a}_l^{(t-1)}. \quad (8)$$

Thus, after the server performs these personalized aggregations for both $\theta$ and $\boldsymbol{a}$, it transmits $(\theta_k^{(t)}, \boldsymbol{a}_k^{(t)})$ back to client $G_k$ for the $(t+1)$-th communication round starting point. Appendix B shows the pseudo code of FedAux.

**Complexity Analysis** For the client side of `FedAux`, the local GNN embedding generation incurs a complexity of $\mathcal{O}(|E_k|d')$, and the auxiliary projection from embeddings to the `APV` results in $\mathcal{O}(N_k d')$. Besides, the kernel-based embedding aggregation over the 1D space induced by the `APV` has complexity $\mathcal{O}(N_k^2 d')$. Consequently, the per-client complexity is $\mathcal{O}(|E_k|d' + N_k^2 d')$. On the server side, computing the client-wise similarity for personalized federated aggregation involves a complexity $\mathcal{O}(K^2 d')$. Therefore, the total complexity of `FedAux` per communication round is $\mathcal{O}\left((|E_k| + N_k^2 + K^2)d'\right)$.

### 3.3 Theoretical Analysis

For notational simplicity, we focus on a single client with $N$ nodes in a given communication round and omit the subscript $k$ and superscript $(t-1)$. The core of our model is to use a learnable auxiliary projection vector `APV` $a$ to capture an optimal node sorting of the local node embeddings and thus serves as a compact summary of the subgraph. However, there is a foundational question that inevitably arises once we replace hard sorting with the smooth kernel aggregator: when the `APV` $a$ is learned via back-propagation, *what does it actually learn? Does it encode an arbitrary nuisance direction, or does it converge to a geometrically meaningful axis that faithfully summarizes the local subgraph?* To answer these questions, we analyze the fidelity of the `APV` with the following theorem.

**Theorem 3.1** (Fidelity of the `APV` $a$). *Let $\mathbf{C} := \frac{1}{N}\sum_{i=1}^{N} h_i h_i^\top$ be the empirical covariance of node embeddings in the current client with size $N$. The gradient of the local loss $\mathcal{L}$ w.r.t. the `APV` $a$ satisfies:*

$$\nabla_a \mathcal{L} = -\frac{2}{\sigma^2}\mathbf{C}a + \mathcal{R}(\sigma), \tag{9}$$

*where the remainder term obeys $\|\mathcal{R}(\sigma)\| = \mathcal{O}(\sigma^0)$ as $\sigma \to 0^+$. Define $\mathbb{S}^{d-1} = \{x \in \mathbb{R}^d : \|x\|_2 = 1\}$ as the unit Euclidean sphere embedded in $\mathbb{R}^d$, then the gradient descent on $\mathcal{L}$ with unit-norm re-normalization reproduces Oja learning rule [22]:*

$$a \leftarrow \Pi_{\mathbb{S}^{d-1}}(a - \eta\mathbf{C}a), \tag{10}$$

*whose unique stable fixed points are the eigenvectors of $\mathbf{C}$, and the global attractor is the principal eigenvector (largest eigenvalue).*

The proof and more discussions are provided in Appendix C.1. It guarantees that, once the kernel aggregator makes $a$ differentiable, ordinary back-propagation forces `APV` to align with the direction along with the node embeddings in that client vary the most. Equivalently, the `APV` $a$ is not an arbitrary trainable knob but a statistically optimal, variance-maximizing summary of local embeddings. Thus, **the `APV` $a$ is provably the first principal component of the local embeddings**.

In Section 3.1, we propose a continuous kernel aggregator to replace the hard sort-then-$\mathrm{Conv1D}$ pipeline used in earlier work [18]. To justify that replacement, the following theorem rigorously shows that the new smooth operator degenerates to the old one in the appropriate limit.

**Theorem 3.2** (Sorting limit and equivalence to $\mathrm{Conv1D}$). *Let $z_i$ be the kernel-smoothed embeddings, and gather them in score order $\widetilde{\mathbf{Z}} = \left[z_{\pi(1)}, \cdots, z_{\pi(N)}\right] \in \mathbb{R}^{N \times d'}$. The original sorted embeddings $\widetilde{\mathbf{H}} = [h_{\pi(1)}, \cdots, h_{\pi(N)}]$ is defined in Eq. (3). Let $\mathbf{W} \in \mathbb{R}^{B \times d'}$ be an arbitrary fixed $\mathrm{Conv1D}$ kernel with zero padding, and denote $\mathrm{Conv}_{\mathbf{W}}(\mathcal{X})_t = \sum_{\tau=1}^{B} \mathbf{W}_\tau \mathcal{X}_{t+\tau-\lceil B/2 \rceil}$, for any sequence $\mathcal{X} \in \mathbb{R}^{N \times d'}$. Then we have:*

$$\lim_{\sigma \to 0^+} \left\|\mathrm{Conv}_{\mathbf{W}}\left(\widetilde{\mathbf{Z}}\right) - \mathrm{Conv}_{\mathbf{W}}\left(\widetilde{\mathbf{H}}\right)\right\|_F = 0, \tag{11}$$

*where $\|\cdot\|_F$ is the Frobenius norm.*

The proof is in Appendix C.2. Theorem 3.2 indicates that the kernel aggregation followed by $\mathrm{Conv1D}$ converges to hard-sorting followed by $\mathrm{Conv1D}$ as the bandwidth $\sigma$ tends to 0. Hence, the two architectures have identical expressive power up to an arbitrarily small error for sufficiently small $\sigma$. Although the limit $\sigma \to 0^+$ recovers discrete sorting, a larger $\sigma$ performs a soft neighborhood pooling that can act as a learnable regularizer against over-fitting noisy local orderings. In practice, we find $\sigma = 1$ to be effective across all datasets.

Next, we present a theoretical analysis of `FedAux`'s convergence rate, which guarantees that it cannot diverge in expectation. Since this analysis focuses on the global model, we use the subscript $\cdot_k$ to denote the client index.

Table 1: Federated node classification results. The reported results are the mean and standard deviation over three different runs. Best performance is highlighted in **bold**.

| | Cora | | | CiteSeer | | | Pubmed | | |
|---|---|---|---|---|---|---|---|---|---|
| **Methods** | **5 Clients** | **10 Clients** | **20 Clients** | **5 Clients** | **10 Clients** | **20 Clients** | **5 Clients** | **10 Clients** | **20 Clients** |
| Local | $81.30_{\pm0.21}$ | $79.94_{\pm0.24}$ | $80.30_{\pm0.25}$ | $69.02_{\pm0.05}$ | $67.82_{\pm0.13}$ | $65.98_{\pm0.17}$ | $84.04_{\pm0.18}$ | $82.81_{\pm0.39}$ | $82.65_{\pm0.03}$ |
| FedAvg | $74.45_{\pm5.64}$ | $69.19_{\pm0.67}$ | $69.50_{\pm3.58}$ | $71.06_{\pm0.60}$ | $63.61_{\pm3.59}$ | $64.68_{\pm1.83}$ | $79.40_{\pm0.11}$ | $82.71_{\pm0.29}$ | $80.97_{\pm0.26}$ |
| FedProx | $72.03_{\pm4.56}$ | $60.18_{\pm7.04}$ | $48.22_{\pm6.81}$ | $71.73_{\pm1.11}$ | $63.33_{\pm3.25}$ | $64.85_{\pm1.35}$ | $79.45_{\pm0.25}$ | $82.55_{\pm0.24}$ | $80.50_{\pm0.25}$ |
| FedPer | $81.68_{\pm0.40}$ | $79.35_{\pm0.04}$ | $78.01_{\pm0.32}$ | $70.41_{\pm0.32}$ | $70.53_{\pm0.28}$ | $66.64_{\pm0.27}$ | $85.80_{\pm0.21}$ | $84.20_{\pm0.28}$ | $84.72_{\pm0.31}$ |
| GCFL | $81.47_{\pm0.65}$ | $78.66_{\pm0.27}$ | $79.21_{\pm0.70}$ | $70.34_{\pm0.57}$ | $69.01_{\pm0.12}$ | $66.33_{\pm0.05}$ | $85.14_{\pm0.33}$ | $84.18_{\pm0.19}$ | $83.94_{\pm0.36}$ |
| FedGNN | $81.51_{\pm0.68}$ | $70.12_{\pm0.99}$ | $70.10_{\pm3.52}$ | $69.06_{\pm0.92}$ | $55.52_{\pm3.17}$ | $52.23_{\pm6.00}$ | $79.52_{\pm0.23}$ | $83.25_{\pm0.45}$ | $81.61_{\pm0.59}$ |
| FedGTA | $71.26_{\pm2.93}$ | $68.33_{\pm1.27}$ | $69.24_{\pm0.91}$ | $69.39_{\pm0.75}$ | $67.34_{\pm1.08}$ | $65.29_{\pm1.92}$ | $78.47_{\pm0.25}$ | $82.79_{\pm0.20}$ | $81.92_{\pm0.60}$ |
| FedSage+ | $72.97_{\pm5.94}$ | $69.05_{\pm1.59}$ | $57.97_{\pm12.6}$ | $70.74_{\pm0.69}$ | $65.63_{\pm3.10}$ | $65.46_{\pm0.74}$ | $79.57_{\pm0.24}$ | $82.62_{\pm0.31}$ | $80.82_{\pm0.25}$ |
| FED-PUB | $83.72_{\pm0.18}$ | $81.45_{\pm0.12}$ | $81.10_{\pm0.64}$ | $72.40_{\pm0.26}$ | $71.83_{\pm0.61}$ | $66.89_{\pm0.14}$ | $86.81_{\pm0.12}$ | $86.09_{\pm0.17}$ | $84.66_{\pm0.54}$ |
| FedAux | $\mathbf{84.57_{\pm0.39}}$ | $\mathbf{82.05_{\pm0.71}}$ | $\mathbf{81.60_{\pm0.64}}$ | $\mathbf{72.99_{\pm0.82}}$ | $\mathbf{73.16_{\pm0.29}}$ | $\mathbf{68.10_{\pm0.35}}$ | $\mathbf{88.10_{\pm0.16}}$ | $\mathbf{86.43_{\pm0.20}}$ | $\mathbf{84.87_{\pm0.42}}$ |

| | Amazon-Computer | | | Amazon-Photo | | | ogbn-arxiv | | |
|---|---|---|---|---|---|---|---|---|---|
| **Methods** | **5 Clients** | **10 Clients** | **20 Clients** | **5 Clients** | **10 Clients** | **20 Clients** | **5 Clients** | **10 Clients** | **20 Clients** |
| Local | $89.22_{\pm0.13}$ | $88.91_{\pm0.17}$ | $89.52_{\pm0.20}$ | $91.67_{\pm0.09}$ | $91.80_{\pm0.02}$ | $90.47_{\pm0.15}$ | $66.76_{\pm0.07}$ | $64.92_{\pm0.09}$ | $65.06_{\pm0.05}$ |
| FedAvg | $84.88_{\pm1.96}$ | $79.54_{\pm0.23}$ | $74.79_{\pm0.24}$ | $89.89_{\pm0.83}$ | $83.15_{\pm3.71}$ | $81.35_{\pm1.04}$ | $65.54_{\pm0.07}$ | $64.44_{\pm0.10}$ | $63.24_{\pm0.13}$ |
| FedProx | $85.25_{\pm1.27}$ | $83.81_{\pm1.09}$ | $73.05_{\pm1.30}$ | $90.38_{\pm0.48}$ | $80.92_{\pm4.64}$ | $82.32_{\pm0.29}$ | $65.21_{\pm0.20}$ | $64.37_{\pm0.18}$ | $63.03_{\pm0.04}$ |
| FedPer | $89.67_{\pm0.34}$ | $89.73_{\pm0.04}$ | $87.86_{\pm0.43}$ | $91.44_{\pm0.37}$ | $91.76_{\pm0.25}$ | $90.59_{\pm0.06}$ | $66.87_{\pm0.05}$ | $64.99_{\pm0.18}$ | $64.66_{\pm0.11}$ |
| GCFL | $89.07_{\pm0.91}$ | $90.03_{\pm0.16}$ | $\mathbf{89.08_{\pm0.25}}$ | $91.99_{\pm0.29}$ | $92.06_{\pm0.25}$ | $90.79_{\pm0.17}$ | $66.80_{\pm0.12}$ | $65.09_{\pm0.08}$ | $65.08_{\pm0.04}$ |
| FedGNN | $88.08_{\pm0.15}$ | $88.18_{\pm0.41}$ | $83.16_{\pm0.13}$ | $90.25_{\pm0.70}$ | $87.12_{\pm2.01}$ | $81.00_{\pm4.48}$ | $65.47_{\pm0.22}$ | $64.21_{\pm0.32}$ | $63.80_{\pm0.05}$ |
| FedGTA | $85.06_{\pm0.82}$ | $84.27_{\pm0.71}$ | $79.46_{\pm0.28}$ | $89.70_{\pm0.67}$ | $76.53_{\pm3.21}$ | $82.02_{\pm0.78}$ | $65.42_{\pm0.09}$ | $64.22_{\pm0.08}$ | $63.75_{\pm0.18}$ |
| FedSage+ | $85.04_{\pm0.61}$ | $80.50_{\pm1.30}$ | $70.42_{\pm0.85}$ | $90.77_{\pm0.44}$ | $76.81_{\pm8.24}$ | $80.58_{\pm1.15}$ | $65.69_{\pm0.09}$ | $64.52_{\pm0.14}$ | $63.31_{\pm0.20}$ |
| FED-PUB | $90.25_{\pm0.07}$ | $89.73_{\pm0.16}$ | $88.20_{\pm0.18}$ | $93.20_{\pm0.15}$ | $\mathbf{92.46_{\pm0.19}}$ | $90.59_{\pm0.35}$ | $67.62_{\pm0.11}$ | $66.35_{\pm0.16}$ | $63.90_{\pm0.27}$ |
| FedAux | $\mathbf{90.38_{\pm0.08}}$ | $\mathbf{89.92_{\pm0.15}}$ | $88.35_{\pm0.96}$ | $\mathbf{93.37_{\pm0.26}}$ | $92.30_{\pm0.29}$ | $\mathbf{90.91_{\pm0.60}}$ | $\mathbf{68.83_{\pm0.15}}$ | $\mathbf{68.50_{\pm0.27}}$ | $\mathbf{65.52_{\pm0.10}}$ |

**Theorem 3.3** (Global linear convergence). *Let $\Psi_k^{(t)} = (\theta_k^{(t)}, \boldsymbol{a}_k^{(t)})$ be the local parameters of client $G_k$ at the communication round t, and $\Psi^{(t)} = \left[\Psi_1^{(t)}, \cdots, \Psi_k^{(t)}\right]$ collects all local parameters. Assuming 1) every local objective is $\mathscr{L}$-smooth: $\forall \Psi_k, \Psi_k' : \|\nabla\mathcal{L}_k(\Psi_k) - \nabla\mathcal{L}_k(\Psi_k')\| \leq \mathscr{L}\|\Psi_k - \Psi_k'\|$; 2) the stochastic gradients are unbiased ($\mathbb{E}[g_k] = \nabla\mathcal{L}_k$) and variance-bounded ($\mathbb{E}\left[\|g_k - \nabla\mathcal{L}_k\|^2\right] \leq \zeta^2$), where $g_k := \nabla_{(\Psi_k)}\mathcal{L}_k$ means the local gradients; 3) each local objective satisfies the $\mu$-PL condition [23]; 4) let $\Omega^{(t)} = [w_{kl}^{(t)}]_{k,l} \in \mathbb{R}^{K \times K}$, the spectral gap $\rho := \sup_t \left\|\Omega^{(t)} - \frac{1}{K}\mathbf{1}\mathbf{1}^\top\right\|_2 < 1$. Let each client perform Q local updates per round, and the learning rate $0 < \eta \leq \frac{1}{2\mathscr{L}}$. With any initial parameters $\Psi^{(0)} = (\theta^{(0)}, \boldsymbol{a}^{(0)})$, we have:*

$$\mathbb{E}\left[\mathcal{L}\left(\Psi^{(T)}\right) - \mathcal{L}^\star\right] \leq (1 - \eta\mu)^{QT}\left(\mathcal{L}\left(\Psi^{(0)}\right) - \mathcal{L}^\star\right) + \frac{\eta\mathscr{L}\zeta^2}{2\mu} + \frac{2\eta\mathscr{L}\rho^2}{\mu(1-\rho)^2}, \qquad (12)$$

*where $\mathcal{L}(\Psi) := \sum_{k=1}^{K} p_k \mathcal{L}_k(\Psi_k)$ is the global objective with client sampling probability $p_k$ (w.l.o.g., $p_k = 1/K$). $\mathcal{L}^\star := \sum_k p_k \mathcal{L}_k^\star$ is the weighted optimal value.*

The proof is in Appendix C.3. In Theorem 3.3, the first term decays linearly; the second is the classical SGD variance term; the third is the personalization error and vanishes as $\rho \to 0$. Thus FedAux can linearly descend to a neighborhood of the global optimum.

## 4 Experiments

### 4.1 Experimental Setup

**Datasets and Experimental Settings** Following previous works [40, 3, 42], we construct distributed subgraphs from benchmark datasets by partitioning each original graph into multiple subgraphs corresponding to individual clients. Specifically, we perform experiments on six widely used datasets, including four citation networks (Cora, CiteSeer, Pubmed [25], and ogbn-arxiv [10]) and two product co-purchase networks (Amazon-Computer and Amazon-Photo [19, 26]). We employ METIS [12] as our default graph partitioning algorithm, which allows explicit specification of the desired number of subgraphs without overlapping. Based on these datasets, we follow the standard experimental setup used in personalized subgraph federated learning literature [3, 42]. Specifically,

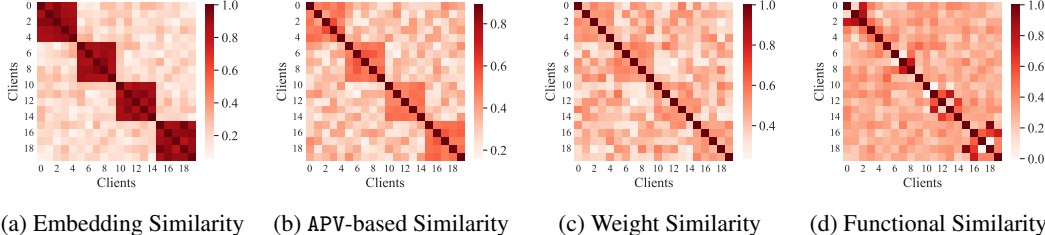

| (a) Embedding Similarity | (b) APV-based Similarity | (c) Weight Similarity | (d) Functional Similarity |

Figure 3: Client similarity based on different measures. Darker colors indicate higher similarity.

for dataset splitting, we randomly sample 20%/40%/40% of nodes from each subgraph for training, validation, and testing, respectively. The only exception is the ogbn-arxiv dataset, due to its significantly larger size. For this dataset, we randomly select 5% of the nodes for training, half of the remaining nodes for validation, and the rest for testing. Dataset statistics and implementation details can be found in Appendix D.

**Baselines** FedAux is compared against several representative federated learning (FL) methods, including general FL baselines: FedAvg [20], FedProx [14], and FedPer [2]; as well as specialized graph-based FL models [1]: GCFL [35], FedGNN [34], FedGTA [16], FedSage+[40], and FED-PUB[3]. Additionally, we consider a local variant of our model (*Local*), where FedAux is trained independently at each client without parameter sharing.

## 4.2 Main Results

As summarized in Table 1, FedAux is the only algorithm that wins every dataset–client-count combination, indicating that the APV-driven personalization generalizes from small citation graphs (Cora, CiteSeer) to large-scale, high-dimensional graphs (ogbn-arxiv). Specifically, relative to the strongest competitor in each column, FedAux achieves accuracy improvements ranging from 0.2% to 2.4%. The margin over the canonical FedAvg is more pronounced, with an average gain of 4.5%, underscoring that the proposed auxiliary projection mechanism confers benefits well beyond classical

Table 2: Degree of non-IIDness. Pubmed exhibits the lowest non-IIDness, and Amazon-Photo has the highest.

| Non-IIDness | **Pubmed** | | |
| | 5 Clients | 10 Clients | 20 Clients |
|---|---|---|---|
| $\xi$ | 0.1316 | 0.1500 | 0.1725 |
| Non-IIDness | **Amazon-Photo** | | |
| | 5 Clients | 10 Clients | 20 Clients |
| $\xi$ | 0.3398 | 0.3668 | 0.4307 |

weighted averaging. Further, to quantify statistical heterogeneity (i.e., degree of non-IIDness), we adopt $\xi = \text{JSD} + \text{MMD}$ where the Jensen–Shannon Divergence (JSD) captures label-distribution skew and the Maximum Mean Discrepancy (MMD) captures disparities in subgraph structure (formal definition in Appendix D.3). Higher values of $\xi$ indicate stronger non-IIDness. As reported in Table 5 of Appendix D.3, $\xi$ rises monotonically for every dataset as the federation enlarges from 5 to 20 clients, confirming that our partition protocol indeed induces progressively harsher heterogeneity. This trend is mirrored in the performance of all methods, whose accuracies decline with larger client counts. Nevertheless, the drop for FedAux is consistently the smallest: on Cora, accuracy falls by only 2.9%, while FedAvg and FedProx lose nearly 7%. To highlight the contrast, we single out the most IID dataset Pubmed and the most non-IID dataset Amazon-Photo in Table 2. The results under these settings show that FedAux remains the top performer in both extremes, indicating that our model is merely tuned for gentle partitions but retains its edge under pronounced non-IID conditions.

## 4.3 Model Analysis

**Effectiveness of APV-based Client Similarity Estimation** To intuitively show that the auxiliary projection vector APV can accurately capture the latent similarity among clients under non-IIDness, we construct a synthetic graph (See Appendix D.4) that jointly embodies the two types of heterogeneity:

---

[1] We exclude weakly privacy-protected baselines such as FedGCN [37], GraphFL [32], and FedStar [30] as these methods leak node embeddings, connectivities, or local gradients.

Table 3: Attack AUC of MIA.

| Dataset | FedGNN | FedGTA | FedAux |
|---------|--------|--------|--------|
| Cora | 0.56 | 0.54 | **0.51** |
| Pubmed | 0.58 | 0.55 | **0.52** |
| ogbn-arxiv | 0.55 | 0.53 | **0.49** |

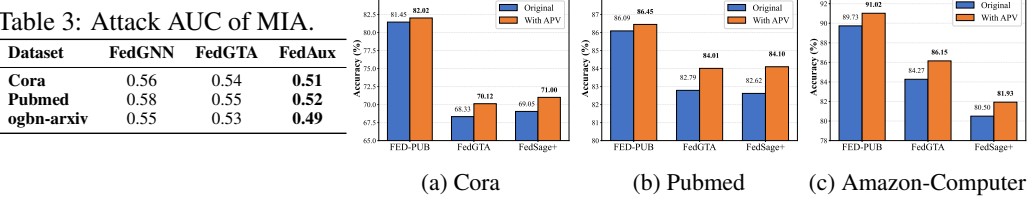

(a) Cora      (b) Pubmed      (c) Amazon-Computer

Figure 4: Transferability of APV.

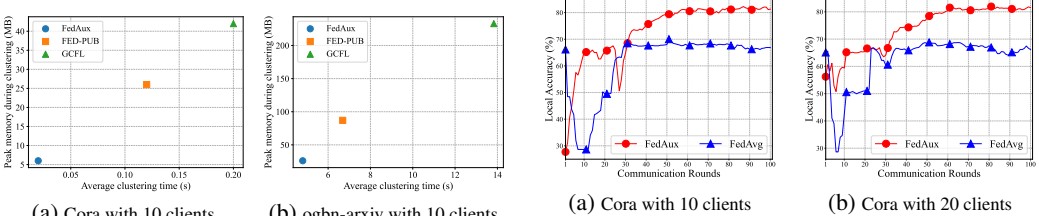

(a) Cora with 10 clients    (b) ogbn-arxiv with 10 clients

Figure 5: Server-side clustering cost.

(a) Cora with 10 clients    (b) Cora with 20 clients

Figure 6: Averaged local accuracy concerning communication rounds.

label heterogeneity and subgraph (structural and feature) heterogeneity. Results in Fig. 3 demonstrate that the APV-based client similarity can best recover the ground-truth client relationships. Each APV converges to the principal axis of its client's embedding distribution, yielding highly aligned directions within the same group and near-orthogonal directions across different groups (Theorem 3.1). Thus APV can serve as a privacy-preserving yet information-rich descriptor in subgraph FL.

**Membership Inference Attacks (MIA)** We add a comprehensive empirical privacy evaluation via MIAs to rigorously test whether our APV-based framework leaks sensitive information. We assume an honest-but-curious server with full access to the entire history of a client's APV $a_k$ and its GNN parameters $\theta_k$. The goal of the server is to decide whether a probe embedding $h_{\text{prob}}$ originated from client $G_k$, using only $(a_k, \theta_k)$, while raw data, node embeddings, and gradients are never shared. Following the standard MIA methodology [21, 27], the server trains an attack classifier $g_k(a_k, \theta_k, h_{\text{prob}})$ implemented as a two-layer MLP with hidden size 64, to output the probability that $h_{\text{prob}}$ belongs to the training set of $G_k$. The attack is trained by maximum likelihood on a held-out mixture of member against non-member probes. In Table 3, we compare FedAux against representative subgraph FL baselines on three datasets using identical client partitions to those in our main experiments, and report the Attack AUC (lower values indicate better privacy) averaged over five random seeds. The results show that MIAs against FedAux achieve AUCs in the range $[0.49, 0.52]$, which is indistinguishable from random guessing. This demonstrates that APVs do not leak sensitive membership information and actually provide stronger privacy than baselines.

**Transferability of APV** To examine whether APV can generalize beyond FedAux and serve as a plug-and-play personalization module, we integrate it into representative subgraph FL baselines. Specifically, we replace FED-PUB's functional embedding similarity with APV, and jointly train APV with FedGTA and FedSage+. All variants are trained under the same 10-client non-IID split on three datasets. Fig. 4 shows that APV consistently improves each baseline across datasets. For example, FedGTA with APV achieves 1%-2% higher accuracy than its original counterpart. These improvements confirm that APV functions as a generic, lightweight, and privacy-preserving similarity proxy that can be seamlessly integrated with diverse subgraph FL frameworks.

**Clustering Efficiency on Server** To perform personalized FL, FED-PUB uses soft clustering by running a proxy graph through each client model and comparing embeddings. GCFL applies hard clustering, grouping clients with a Stoer-Wagner cut on cross-round gradients. Our APV-based method only computes similarity between the uploaded vectors, requiring no extra forward passes, gradient logging or cut computation, so server overhead is minimal. As Fig. 5 illustrates, the APV method attains the fastest clustering time and the lowest peak memory, making it more efficient at scale.

**Convergence Rates** Fig. 6 compares the convergence behavior of `FedAux` and FedAvg. Notably, `FedAux` converges by the 60th communication round in both the 10-client and 20-client settings. Since the latter involves a higher level of data heterogeneity, it indicates that `FedAux` maintains a consistent convergence speed even as the degree of non-IIDness increases.

In Appendix E, we conduct an ablation study to investigate the impact of our proposed continuous aggregation scheme, which encodes local node relationships in the 1D space induced by the `APV`, as well as the effect of the server-side `APV`-based personalized federated aggregation. We also conduct experiments to analyze the sensitivity to hyperparameters.

## 5    Related Work

For general **Federated Learning**, FedAvg [20] first demonstrated that deep models can be trained on decentralized data with iterative model averaging. Subsequent work revealed that statistical and systems heterogeneity slow or destabilize FedAvg's convergence, which has been formally addressed by proximal correction [14], variance reduction [11], and data–distribution smoothing through a small globally shared subset of samples [43]. Optimization refinements such as normalized aggregation [33], adaptive server updates [24] and dynamic regularization [1] further tighten convergence guarantees under extreme non-IID settings. Beyond a single global model, personalization frameworks, e.g., Ditto [15], which jointly optimizes a shared model and client-specific objectives, explicitly trade off fairness, robustness, and local adaptation. These advances establish the algorithmic and theoretical foundations on which federated graph learning is built.

For **Graph Federated Learning** [36, 17, 39, 38], at the node level, FedGCN [37] illustrates that one-shot encrypted exchange can suffice to federate GCNs while maintaining accuracy and privacy; GCFL [35] clusters clients by gradient dynamics to mitigate structural and feature shift across graphs. To combat cross-domain heterogeneity, FedStar [30] extracts a domain-invariant topology that generalizes across diverse graphs, and FedGraph [9] augments local data by requesting node information from other clients. When each participant owns only a fragment of a larger network, subgraph federated learning methods such as FedSage/FedSage+ [40] generate virtual neighbours to repair missing cross-subgraph edges. FED-PUB [3] proposes to generate functional embeddings to evaluate the similarity between clients for personalized aggregation. These models collectively highlight an open challenge for personalized graph FL, i.e., accurate client similarity measures, which our proposed `FedAux` addresses through end-to-end learning of auxiliary projection vectors.

Several new methods further extend personalization but come with non-trivial trade-offs. FedSSP [31] transmits spectral components that are invariant across domains but can leak sensitive local spectral information. FedEgo [41] shares ego-network embeddings, which directly expose structural patterns. PFGNAS [8] leverages prompt-based neural architecture search without server-side similarity estimation, but its reliance on LLM-based personalization introduces prohibitive cost. FedGrAINS [5] personalizes within each client by learning with a GFlowNet, thereby avoiding server-side mixing, but at the expense of training an additional model with high compute and memory overhead. While the aforementioned methods contribute interesting ideas, they either compromise privacy guarantees (e.g., FedSSP, FedEgo) or impose heavy computational costs (e.g., PFGNAS, FedGrAINS), limiting their practicality. In contrast, our method learns Auxiliary Projection Vectors (APV) in an end-to-end manner: lightweight, privacy-preserving client signatures that avoid raw data or embedding leakage, yet enable effective similarity-aware aggregation for personalized subgraph FL.

## 6    Conclusion

We present `FedAux`, a personalized subgraph federated learning framework that augments each local GNN with learnable auxiliary projection vectors (`APVs`). Specifically, besides the global GNN parameters, the server initializes and distributes `APVs` to each client, enabling effective and privacy-preserving characterization of local subgraph structures. By continuously projecting node embeddings onto a 1D space induced by these `APVs`, local models adaptively refine the `APVs` to optimally capture node relationships within each client's subgraph. After each communication round, the server leverages similarity between client `APVs` to perform personalized model aggregation. Extensive experiments across multiple datasets with varying numbers of clients validate the effectiveness of `APVs` as informative descriptors for personalized subgraph FL.

## Acknowledgements

The research is supported, in part, by the Ministry of Education, Singapore, under its Academic Research Fund Tier 1 (RG101/24).

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

# A    More Discussion of Related Work

In subgraph federated learning, each client holds a local subgraph $G_k$ of a global graph $G$. These subgraphs can vary substantially in their feature distributions, structural/topological properties, and label distributions. Thus, simply applying FedAvg may fail to converge properly or yield a suboptimal global model, because clients often learn different parameters tailored to their local subgraphs, and blindly averaging those parameters disregards the non-IID nature of the data. To address this, recent studies move beyond basic FedAvg by introducing personalized or locality-aware aggregation schemes that better handle heterogeneous subgraph data.

GCFL [35] compares each client's parameter updates before and after communication. If a client's update deviates substantially from the majority, it is deemed too different and is excluded from the dominant aggregation cluster. However, GCFL relies on hard clustering, which cannot capture finer-grained similarities across clients. Moreover, it depends on manually tuned hyperparameters to control how different a client must be before exclusion, leaving it unclear how large a deviation is tolerable without enough knowledge about the raw data. FED-PUB [3], on the other hand, constructs a proxy graph at the server and evaluates model outputs from each client on the proxy graph to measure their similarity. Clients with similar outputs on the proxy graph are deemed more alike and thus aggregated more closely. Yet building a suitable proxy graph as the middleware to measure the similarity between clients is nontrivial, because it requires simulating the real graph data on the server.

In this work, we propose an orthogonal approach to measure and aggregate models on the server. Specifically, beyond distributing a global GNN, the server also provides a global Auxiliary Projection Vector (APV) to each client at the start of every communication round, as shown on the left of Fig. 1. During local training, the APV is jointly optimized with the client's GNN parameters to form a one-dimensional space onto which node embeddings are projected. This procedure tailors the APV, which adjusts the shape of this space, to the unique distribution of each subgraph by preserving its distinctive patterns. Once training concludes, clients upload their updated GNN parameters and local APV to the server, which then compares these learned APVs to gauge similarity and detect finer-grained, continuous heterogeneity across clients. By discarding the hard clustering thresholds used in GCFL and removing the need for a proxy dataset used in FED-PUB, our method offers a more flexible and efficient strategy to identify and aggregate similar subgraphs.

Another advantage of our method lies in its privacy-preserving design. We argue that sharing detailed node embeddings, as in FedGCN [37], can leak private subgraph information, as adversaries on the server could compute pairwise similarities among embeddings to reconstruct local connectivity. Moreover, due to high dimensionality and lack of explicit structural encoding, directly comparing parameter matrices to measure raw data similarity is both unreliable and computationally unstable. In contrast, our proposed APV is a compact parameter vector that preserves essential node relationships without exposing the actual node embeddings, offering stronger privacy guarantees and structure-awareness. Additionally, its low-dimensional nature makes similarity computation more efficient and robust.

# B    Pseudo Code of `FedAux`

The overall training algorithm is shown in Algorithm 1.

# C    Proofs

## C.1    Proof of Theorem 3.1

The proof relies on two mild structural assumptions, both typical in optimization analyses of deep models.

**Assumption C.1** (Centred embeddings). $\bar{h} := \frac{1}{N} \sum_i h_i = \mathbf{0}$, *where $N$ is the number of nodes in the current client.*

**Assumption C.2** (Linear classifier Jacobian). *Let $g_i := \nabla_{z_i} \mathcal{L}(\theta, \boldsymbol{a})$. Assuming that the mapping $z_i \mapsto g_i$ is linear: $g_i = \mathbf{W} z_i$ for some matrix $\mathbf{W} \in \mathbb{R}^{d' \times d'}$.*

**Algorithm 1:** FedAux: Subgraph FL with Differentiable Auxiliary Projections

---

**Input:** number of clients $K$; global communication rounds $T$; local steps per round $Q$; learning
          rates $\eta$; similarity temperature $\alpha$

**Init:** server initializes global GNN weights $\theta^{(0)}$ and APV $\boldsymbol{a}^{(0)} \sim \mathcal{N}(0, I_{d'})$; each client $G_k$ sets
          $\theta_k^{(0)} \leftarrow \theta^{(0)}, \boldsymbol{a}_k^{(0)} \leftarrow \boldsymbol{a}^{(0)}$

1 **for** $t \leftarrow 1$ **to** $T$ **do**
    // Server $\longrightarrow$ Clients
2     broadcast $\{(\theta_k^{(t-1)}, \boldsymbol{a}_k^{(t-1)})\}_{k=1}^{K}$ to each client $\{G_k\}_{k=1}^{K}$;
    // Local training on each client $G_k$ (runs in parallel)
3     **for** *each client* $G_k \in \{G_1, \ldots, G_K\}$ **do in parallel**
4         $(\theta_k, \boldsymbol{a}_k) \leftarrow (\theta_k^{(t-1)}, \boldsymbol{a}_k^{(t-1)})$;
5         **for** $q \leftarrow 1$ **to** $Q$ **do**
6             run local GNN forward pass with $h_{k,i}^{(t-1)} = f_{\theta_k^{(t-1)}}(v_i)$ to learn node embeddings as
              Eq. (1);
7             compute similarity scores $s_k^{(t-1)}$ with $s_{k,i}^{(t-1)} = \langle \hat{h}_{k,i}^{(t-1)}, \boldsymbol{a}_k^{(t-1)} \rangle$;
8             compute kernel weights $\kappa(s_{k,i}^{(t-1)}, s_{k,j}^{(t-1)})$;
9             compute kernel-based aggregated embeddings $z_{k,i}^{(t-1)}$ with Eq. (6);
10            concatenate $r_{k,i}^{(t-1)} = [h_{k,i}^{(t-1)} \| z_{k,i}^{(t-1)}]$;
11            forward through the classifier; compute loss $\mathcal{L}_{CE}$;
12            update $(\theta_k^{(t-1)}, \boldsymbol{a}_k^{(t-1)})$ w.r.t. $\mathcal{L}_{CE}$ with learning rate $\eta$;
13         **end**
14         upload $(\theta_k^{(t-1)}, \boldsymbol{a}_k^{(t-1)})$ to server;
15     **end**
    // Server-side personalised aggregation
16     compute $w_{k,l}^{(t-1)}$ with Eq. (7);
17     **for** *each client* $G_k$ **do**
18         $\theta_k^{(t)} \leftarrow \sum_{l=1}^{K} w_{k,l}^{(t-1)} \theta_l^{(t-1)}$;
19         $\boldsymbol{a}_k^{(t)} \leftarrow \sum_{l=1}^{K} w_{k,l}^{(t-1)} \boldsymbol{a}_l^{(t-1)}$;
20     **end**
21     send $(\theta_k^{(t)}, \boldsymbol{a}_k^{(t)})$ back to client $G_k$;
22 **end**

**Output:** personalized models $\left\{\theta_k^{(T)}, \mathbf{a}_k^{(T)}\right\}_{k=1}^{K}$ for $K$ clients

---

Theorem C.2 holds exactly for a linear-softmax classifier and is a first-order approximation for MLPs.
Then we introduce the auxiliary lemmata.

**Lemma C.3** (Gradient of similarity score). *For node* $v_i$, $\frac{\partial s_i}{\partial \boldsymbol{a}} = h_i - s_i \boldsymbol{a}$ *holds.*

*Proof.* $s_i = \boldsymbol{a}^\top h_i$, with $\|\boldsymbol{a}\| = 1$. Hence $\frac{\partial s_i}{\partial \boldsymbol{a}} = h_i$. Because we will always re-project $\boldsymbol{a}$ on the unit
sphere after every update, the tangential component $h_i - s_i \boldsymbol{a}$ is the effective gradient, and the radial
component vanishes. $\square$

**Lemma C.4** (Gradient of kernel entry). $\frac{\partial \mathcal{K}_{ij}}{\partial \boldsymbol{a}} = -\frac{2}{\sigma^2} (s_i - s_j) \mathcal{K}_{ij} [(h_i - h_j) - (s_i - s_j) \boldsymbol{a}]$.

*Proof.* Apply the chain rule to $\mathcal{K}_{ij} = \kappa(s_i, s_j) = \exp\left(-\frac{1}{\sigma^2} (s_i - s_j)^2\right)$ and invoke Theorem C.3
for $\partial(s_i - s_j)/\partial \boldsymbol{a}$. $\square$

**Lemma C.5** (Gradient of the kernel-smoothed embedding). *Define the normalized kernel weights:*

$$\beta_{ij} := \frac{\mathcal{K}_{ij}}{M_i}, \quad \sum_j \beta_{ij} = 1. \tag{13}$$

*Then we have*

$$\frac{\partial z_i}{\partial \boldsymbol{a}} = -\frac{2}{\sigma^2} \sum_{j=1}^{N} \beta_{ij} \left(s_i - s_j\right) \left(h_j - z_i\right) \otimes \left[\left(h_i - h_j\right) - \left(s_i - s_j\right) \boldsymbol{a}\right], \tag{14}$$

*where $\otimes$ denotes the outer product.*

*Proof.* Let $D_i = \sum_{j=1}^{N} \mathcal{K}_{ij} h_j$, based on Eq. (6) we have $z_i = \frac{D_i}{M_i}$. Using the quotient rule, we have:

$$\frac{\partial z_i}{\partial \boldsymbol{a}} = \frac{1}{M_i} \frac{\partial D_i}{\partial \boldsymbol{a}} - \frac{D_i}{M_i^2} \frac{\partial M_i}{\partial \boldsymbol{a}}. \tag{15}$$

Then we compute the two gradients in Eq. (15). The gradient of the $D_i$ w.r.t. $\boldsymbol{a}$ is $\frac{\partial D_i}{\partial \boldsymbol{a}} = \sum_{j=1}^{N} \frac{\partial \mathcal{K}_{ij}}{\partial \boldsymbol{a}} h_j^\top$, where $\frac{\partial \mathcal{K}_{ij}}{\partial \boldsymbol{a}}$ has been given by Theorem C.4. Thus we have:

$$\frac{\partial D_i}{\partial \boldsymbol{a}} = -\frac{2}{\sigma^2} \sum_{j=1}^{N} \left(s_i - s_j\right) \mathcal{K}_{ij} h_j \left[\left(h_i - h_j\right) - \left(s_i - s_j\right) \boldsymbol{a}\right]^\top. \tag{16}$$

The Gradient of $M_i$ can be represented as:

$$\frac{\partial M_i}{\partial \boldsymbol{a}} = \sum_{j=1}^{N} \frac{\partial \mathcal{K}_{ij}}{\partial \boldsymbol{a}} = -\frac{2}{\sigma^2} \sum_{j=1}^{N} \left(s_i - s_j\right) \mathcal{K}_{ij} \left[\left(h_i - h_j\right) - \left(s_i - s_j\right) \boldsymbol{a}\right]. \tag{17}$$

We can substitute Eq. (16) and Eq. (17) into the quotient rule Eq. (15):

$$
\begin{aligned}
\frac{\partial z_i}{\partial \boldsymbol{a}} = {}& -\frac{2}{\sigma^2} \frac{1}{M_i} \sum_{j=1}^{N} \left(s_i - s_j\right) \mathcal{K}_{ij} h_j \left[\left(h_i - h_j\right) - \left(s_i - s_j\right) \boldsymbol{a}\right]^\top \\
& + \frac{2}{\sigma^2} \frac{1}{M_i^2} \left(\sum_{j=1}^{N} \mathcal{K}_{ij} h_j\right) \left(\sum_{\ell=1}^{N} \left(s_i - s_\ell\right) \mathcal{K}_{i\ell} \left[\left(h_i - h_\ell\right) - \left(s_i - s_\ell\right) \boldsymbol{a}\right]^\top\right).
\end{aligned}
\tag{18}
$$

Given Eq. (13), Eq. (18) can be rewritten as:

$$
\begin{aligned}
\frac{\partial z_i}{\partial \boldsymbol{a}} = {}& -\frac{2}{\sigma^2} \sum_{j=1}^{N} \beta_{ij} \left(s_i - s_j\right) h_j \left[\left(h_i - h_j\right) - \left(s_i - s_j\right) \boldsymbol{a}\right]^\top \\
& + \frac{2}{\sigma^2} \underbrace{\left(\sum_{j=1}^{N} \beta_{ij} h_j\right)}_{z_i} \left(\sum_{\ell=1}^{N} \beta_{i\ell} \left(s_i - s_\ell\right) \left[\left(h_i - h_\ell\right) - \left(s_i - s_\ell\right) \boldsymbol{a}\right]^\top\right).
\end{aligned}
\tag{19}
$$

By re-indexing $\ell \to j$, Eq. (19) can be represented as:

$$\frac{\partial z_i}{\partial \boldsymbol{a}} = -\frac{2}{\sigma^2} \sum_{j=1}^{N} \beta_{ij} \left(s_i - s_j\right) \left(h_j - z_i\right) \left[\left(h_i - h_j\right) - \left(s_i - s_j\right) \boldsymbol{a}\right]^\top, \tag{20}$$

which is exactly the Eq. (14), completing the derivation. $\square$

### C.1.1 Proof of Theorem 3.1

*Proof.* Using Theorem C.2 and Theorem C.5, we can perform chain-rule expansion of $\nabla_{\boldsymbol{a}} \mathcal{L}$ as:

$$\nabla_{\boldsymbol{a}} \mathcal{L} = \frac{1}{N} \sum_{i=1}^{N} \left(\frac{\partial z_i}{\partial \boldsymbol{a}}\right)^\top g_i = \frac{1}{N} \sum_{i=1}^{N} \sum_{j=1}^{N} \left(\frac{\partial z_i}{\partial \boldsymbol{a}}\right)^\top \mathbf{W} z_i. \tag{21}$$

Substituting Eq. (14) into Eq. (21) gives:

$$\nabla_{\boldsymbol{a}} \mathcal{L} = -\frac{2}{N\sigma^2} \sum_{i=1}^{N} \sum_{j=1}^{N} \beta_{ij} \left(s_i - s_j\right) \left[\left(h_i - h_j\right) - \left(s_i - s_j\right) \boldsymbol{a}\right] \left(h_j - z_i\right)^\top \mathbf{W} z_i. \tag{22}$$

For small $\sigma$, $\beta_{ij}$ is sharply peaked at $j = i$. Given $\varepsilon_{ij} := s_i - s_j$, since $\beta_{ij} \leq e^{-\varepsilon_i^2 j / \sigma^2}$, all terms with $j \neq i$ are exponentially suppressed, and the dominant contribution arises from the linearization around $\varepsilon_{ij} = 0$. Then we conduct Taylor expansion to first order in $\varepsilon_{ij}$:

$$\nabla_a \mathcal{L} = -\frac{2}{N\sigma^2} \sum_{i=1}^{N} \sum_{j=1}^{N} \left[ \beta_{ij} \varepsilon_{ij} h_i \left( h_i^\top \mathbf{W} h_i \right) \right] + \mathcal{O}\left( \sigma^0 \right). \tag{23}$$

In Big-O notation, the symbol $\mathcal{O}\left( \sigma^\mu \right)$ as $\sigma \to 0^+$ means that there exists a constant $c > 0$ and a neighbourhood $(0, \sigma_0]$ such that $|\mathcal{O}\left( \sigma^\mu \right)| \leq c\sigma^\mu$. Using Theorem C.1 and symmetry of the inner summation one obtains the compact matrix form of the above formula:

$$\nabla_a \mathcal{L} = -\frac{2}{\sigma^2} \left( \frac{1}{N} \sum_{i=1}^{N} h_i h_i^\top \right) a + \mathcal{R}(\sigma) = -\frac{2}{\sigma^2} \mathbf{C} a + \mathcal{R}(\sigma), \tag{24}$$

with $\|\mathcal{R}(\sigma)\| = \mathcal{O}\left( \sigma^0 \right)$. This proves Eq. (9).

Since $a$ is re-normalised after every update, the effective tangential gradient [7] is $\nabla_a \mathcal{L} - (a^\top \nabla_a \mathcal{L})a$. Note that $a^\top \mathbf{C} a$ is scalar, so subtracting the radial part yields the tangential gradient $-\mathbf{C} a + \left( a^\top \mathbf{C} a \right) a$. A projected gradient descent step with learning rate $\eta$ therefore becomes:

$$a \leftarrow \Pi_{\mathbb{S}^{d-1}} \left( a - \eta \left[ \mathbf{C} a - \left( a^\top \mathbf{C} a \right) a \right] \right) = \Pi_{\mathbb{S}^{d-1}}((\mathbf{I} - \eta\mathbf{C})a), \tag{25}$$

which is exactly the discrete-time Oja [22] update Eq. (10).

Let $\lambda_{\max}$ be the largest eigenvalue of $\mathbf{C}$, with unit-norm eigenvector $u_{\max}$. Standard theory of Oja's algorithm [13, 22] states:

- Every eigenvector of $\mathbf{C}$ is a fixed point of Eq. (10).

- All eigenvectors other than $\pm u_{\max}$ are unstable, and $\pm u_{\max}$ are globally asymptotically stable provided $0 < \eta < 2/\lambda_{\max}$.

Hence gradient descent drives $a$ toward $\pm u_{\max}$. Because the kernel and the classifier do not depend on the sign of $a$, both directions are equivalent, and choosing the positive-projection suffices. $\square$

### C.1.2 Interpretation of Theorem 3.1

**Rayleigh-quotient maximization**  Oja learning rule is a stochastic gradient ascent on the Rayleigh quotient $\mathcal{R}(a) = a^\top \mathbf{C} a$ over the unit sphere. Theorem 3.1 therefore formalizes the intuition that the APV aligns with the **direction of maximum embedding variance**.

**Role of the bandwidth $\sigma$**  The leading term Eq. (10) is multiplied by $1/\sigma^2$. A smaller bandwidth increases the gradient magnitude, accelerating alignment but reducing smoothness; conversely, a larger $\sigma$ slows convergence while preserving differentiability.

**Compatibility with the global optimization**  Once the APV converges to $u_{\max}$, the kernel weights $\mathcal{K}_{ij}$ depend only on $\langle h_i, u_{\max} \rangle - \langle h_j, u_{\max} \rangle$, which maximally separates nodes along the most informative one-dimensional projection. This precisely captures the fidelity property we desire.

### C.2 Proof of Theorem 3.2

The proof is based on three lemmata.

**Lemma C.6** (Weight concentration). *For each row $i$ of the kernel matrix $\mathcal{K}$, we have:*

$$\lim_{\sigma \to 0+} \mathcal{K}_{ij} = \delta_{ij}, \tag{26}$$

*where $\delta_{ij}$ is the Kronecker delta.*

*Proof.* Since all scores are distinct, setting $\Delta_i := \min_{j \neq i} |s_i - s_j| > 0$. For $j \neq i$ we have:

$$\frac{\mathcal{K}_{ij}}{\mathcal{K}_{ii}} = \exp\left(-\frac{\left((s_i - s_j)^2 - 0\right)}{\sigma^2}\right) \leq \exp\left(-\frac{\Delta_i^2}{\sigma^2}\right). \tag{27}$$

Hence for all $j \neq i$, $\mathcal{K}_{ij} \leq e^{-\Delta_i^2/\sigma^2} \to 0$ as $\sigma \to 0^+$. Since each row of $\mathcal{K}$ is a probability distribution, the diagonal entry must satisfy $\mathcal{K}_{ii} = 1 - \sum_{j \neq i} \mathcal{K}_{ij} \to 1$. $\qquad\square$

**Lemma C.7** (Pointwise convergence of smoothed embeddings)**.**

$$\lim_{\sigma \to 0^+} z_i = h_i, \quad \forall i = 1, \dots, N \tag{28}$$

*Proof.* We can rewrite $z_i$ as:

$$z_i = \sum_{j=1}^N \alpha_{ij} h_j = \mathcal{K}_{ii} h_i + \sum_{j \neq i} \mathcal{K}_{ij} h_j. \tag{29}$$

By Theorem C.6 the non-diagonal weights vanish and $\mathcal{K}_{ii} \to 1$. Therefore $z_i \to h_i$. $\qquad\square$

**Lemma C.8** (Matrix convergence in Frobenius norm)**.**

$$\lim_{\sigma \to 0^+} \left\|\widetilde{\mathbf{Z}} - \widetilde{\mathbf{H}}\right\|_F = 0. \tag{30}$$

*Proof.* Note that both matrices $\widetilde{\mathbf{Z}}$ and $\widetilde{\mathbf{H}}$ have the same ordering $\pi$ of rows. From Theorem C.7 each corresponding row converges as $\left\|z_{\pi(t)} - h_{\pi(t)}\right\|_2 \to 0$ for every $t$. Since $N$ is finite, the Frobenius norm also converges to 0. $\qquad\square$

### C.2.1 Proof of Theorem 3.2

*Proof.* The $\mathrm{Conv1D}$ operator $\mathrm{Conv}_{\mathbf{W}}$ is linear and its Lipschitz constant with respect to the Frobenius norm is $\mathrm{LIP}_{\mathbf{W}} = \left(\sum_{\tau=1}^B \|\mathbf{W}_\tau\|_2^2\right)^{1/2}$, then for any two sequences $\mathcal{X}$ and $\mathcal{Y}$ we have:

$$\|\mathrm{Conv}_{\mathbf{W}}(\mathcal{X}) - \mathrm{Conv}_{\mathbf{W}}(\mathcal{Y})\|_F \leq \mathrm{LIP}_{\mathbf{W}} \|\mathcal{X} - \mathcal{Y}\|_F. \tag{31}$$

Applying this bound with $\mathcal{X} = \widetilde{\mathbf{Z}}$ and $\mathcal{Y} = \widetilde{\mathbf{H}}$ yields:

$$\left\|\mathrm{Conv}_{\mathbf{W}}(\widetilde{\mathbf{Z}}) - \mathrm{Conv}_{\mathbf{W}}(\widetilde{\mathbf{H}})\right\|_F \leq \mathrm{LIP}_{\mathbf{W}} \|\widetilde{\mathbf{Z}} - \widetilde{\mathbf{H}}\|_F. \tag{32}$$

Theorem C.8 states that the right-hand side of Eq. (32) converges to 0, thus the left-hand side must converge to zero as well, establishing the claimed limit. $\qquad\square$

## C.3 Proof of Theorem 3.3

### C.3.1 Local Training Analysis

For a client $G_k$ at any communication round and inner step $q \in \{1, \cdots, Q\}$, let $\Psi_k^q := (\theta_k^q, \boldsymbol{a}_k^q)$ at any communication round. Given the assumption that the local objective $\mathcal{L}_k$ is differentiable and $\mathscr{L}$-smooth: $\forall \Psi_k, \Psi_k' : \|\nabla \mathcal{L}_k(\Psi_k) - \nabla \mathcal{L}_k(\Psi_k')\| \leq \mathscr{L} \|\Psi_k - \Psi_k'\|$, where the smoothness holds for cross-entropy composed with neural networks whose activations are Lipschitz, we have:

$$\mathcal{L}_k(\Psi_k') \leq \mathcal{L}_k(\Psi_k) + \langle \nabla \mathcal{L}_k(\Psi_k), \Psi_k' - \Psi_k \rangle + \frac{\mathscr{L}}{2} \|\Psi_k' - \Psi_k\|^2, \quad \forall \Psi_k, \Psi_k'. \tag{33}$$

Take $\Psi_k = \Psi_k^{s-1}$, and $\Psi_k' = \Psi_k^q = \Psi_k^{q-1} - \eta g_k^{q-1}$:

$$\mathcal{L}_k(\Psi_k^q) \leq \mathcal{L}_k\left(\Psi_k^{q-1}\right) - \eta \left\langle \nabla \mathcal{L}_k\left(\Psi_k^{q-1}\right), g_k^{q-1} \right\rangle + \frac{\mathscr{L}\eta^2}{2} \left\|g_k^{q-1}\right\|^2. \tag{34}$$

Due to the unbiasedness assumption, i.e., $\mathbb{E}[g_k] = \nabla \mathcal{L}_k$, we have:

$$\mathbb{E}\left[\langle \nabla \mathcal{L}_k, g_k^s \rangle\right] = \langle \nabla \mathcal{L}_k, \mathbb{E}[g_k^q] \rangle = \|\nabla \mathcal{L}_k\|^2. \tag{35}$$

Also by the bounded-variance assumption, i.e., $\mathbb{E}_q\left[\|g_k - \nabla\mathcal{L}_k\|^2\right] \leq \zeta^2$, we have:

$$\mathbb{E}_q\left[\|g_k^q\|^2\right] = \|\nabla\mathcal{L}_k\|^2 + \mathbb{E}\left[\|g_k^q - \nabla\mathcal{L}_k\|^2\right] \overset{\text{Ass. 2}}{\leq} \|\nabla\mathcal{L}_k\|^2 + \zeta^2. \tag{36}$$

Then we can insert Eq. (35) and Eq. (36) into Eq. (34) and take expectation as follows:

$$\begin{aligned}\mathbb{E}_q\left[\mathcal{L}_k\left(\Psi_k^q\right)\right] &\leq \mathcal{L}_k\left(\Psi_k^{q-1}\right) - \eta\left\|\nabla\mathcal{L}_k(\Psi_k^{q-1})\right\|^2 + \frac{\mathscr{L}\eta^2}{2}\left(\left\|\nabla\mathcal{L}_k(\Psi_k^{q-1})\right\|^2 + \zeta^2\right) \\ &= \mathcal{L}_k\left(\Psi_k^{q-1}\right) - \left(\eta - \frac{\mathscr{L}\eta^2}{2}\right)\|\nabla\mathcal{L}_k\|^2 + \frac{\mathscr{L}\eta^2\zeta^2}{2}.\end{aligned} \tag{37}$$

Because $\eta \leq 1/2\mathscr{L}$ we have $1 - \mathscr{L}\eta/2 \geq 1/2$, hence:

$$\mathbb{E}_q\left[\mathcal{L}_k\left(\Psi_k^q\right)\right] \leq \mathcal{L}_k\left(\Psi_k^{q-1}\right) - \frac{\eta}{2}\left\|\nabla\mathcal{L}_k\left(\Psi_k^{q-1}\right)\right\|^2 + \frac{\mathscr{L}\eta^2\zeta^2}{2}. \tag{38}$$

Based on the assumption that each local objective $\mathcal{L}_k$ satisfies the $\mu$-PL [2] (Polyak-Lojasiewicz) condition [23] iff

$$\left\|\nabla\mathcal{L}_k(\Psi_k^{q-1})\right\|^2 \geq 2\mu\left(\mathcal{L}_k(\Psi_k^{q-1}) - \mathcal{L}_k^\star\right). \tag{39}$$

Plug Eq. (39) into Eq. (38):

$$\mathbb{E}_q\left[\mathcal{L}_k\left(\Psi_k^q\right) - \mathcal{L}_k^\star\right] \leq (1 - \eta\mu)\left(\mathcal{L}_k\left(\Psi_k^{q-1}\right) - \mathcal{L}_k^\star\right) + \frac{\mathscr{L}\eta^2\zeta^2}{2}. \tag{40}$$

We define the gap as $\Delta_k^{q-1} := \mathcal{L}_k(\Psi_k^{q-1}) - \mathcal{L}_k^\star$. Taking the total expectation and iterating Eq. (40) $Q$ times, we have:

$$\mathbb{E}\left[\Delta_k^Q\right] \leq (1 - \eta\mu)^Q\Delta_k^1 + \frac{\mathscr{L}\eta^2\zeta^2}{2}\sum_{j=1}^Q(1 - \eta\mu)^{j-1}. \tag{41}$$

The geometric sum is:

$$\sum_{j=1}^Q(1 - \eta\mu)^{j-1} = \frac{1 - (1 - \eta\mu)^Q}{\eta\mu} \leq \frac{1}{\eta\mu}. \tag{42}$$

Therefore, the following inequality holds:

$$\mathbb{E}\left[\Delta_k^Q\right] \leq (1 - \eta\mu)^Q\Delta_k^1 + \frac{\eta\mathscr{L}\zeta^2}{2\mu}, \tag{43}$$

which is the per-client local-training contraction.

### C.3.2 Effect of Global Kernel-Based Aggregation

Define $\Psi_k^{\text{loc},(t-1)} := \Psi_k^Q$ which means the local client parameters after $Q$ local training iterations. Let $\mathbf{f}^{(t-1)} := \left[f_1^{(t-1)}, \cdots, f_K^{(t-1)}\right]^\top$, where $f_k^{(t-1)} = \mathcal{L}_k(\Psi_k^{\text{loc},(t-1)}) - \mathcal{L}_k^\star$. Recall our proposed personalized aggregation scheme in Eq. (8), it can be rewritten as:

$$\Psi_k^{(t)} = \sum_{l=1}^K w_{kl}^{(t-1)}\Psi_l^{\text{loc},(t-1)}. \tag{44}$$

Since each new parameter is a convex combination Eq. (44), based on the Jensen's inequality and $\mathscr{L}$-smoothness assumption in Eq. (33), the following inequality holds:

$$\mathcal{L}_k(\Psi_k^{(t)}) = \sum_{l=1}^K w_{kl}^{(t-1)}\mathcal{L}_k(\Psi_l^{\text{loc},(t-1)}). \tag{45}$$

---

[2]Here, we slightly abuse the notation $\mu$, which was previously introduced in Section C.1.1 with a different meaning.

Let $\mathbf{p} = [p_1, \ldots, p_K]^\top$, Eq. (45) subtracts $\mathcal{L}_k^\star$ and multiply by $p_k$, and sum over $k$, we can reach:

$$\mathcal{L}(\Psi^{(t)}) - \mathcal{L}^\star \le \mathbf{p}^\top \Omega^{(t-1)} \mathbf{f}^{(t-1)}. \tag{46}$$

Let the global average gap as $\bar{f}^{(t-1)} := \mathbf{p}^\top \mathbf{f}^{(t-1)}$ and $\mathbf{r}^{(t-1)} := \mathbf{f}^{(t-1)} - \bar{f}^{(t-1)} \mathbf{1}$. Since row-stochasticity implies $\Omega^{(t-1)} \mathbf{1} = \mathbf{1}$, we have:

$$\mathbf{p}^\top \Omega^{(t-1)} \mathbf{f}^{(t-1)} = \bar{f}^{(t-1)} + \mathbf{p}^\top \Omega^{(t-1)} \mathbf{r}^{(t-1)}. \tag{47}$$

Hence:

$$\mathcal{L}\left(\Psi^{(t)}\right) - \mathcal{L}^\star \le \bar{f}^{(t-1)} + \mathbf{p}^\top \mathbf{V}^{(t-1)} \mathbf{r}^{(t-1)}, \quad \mathbf{V}^{(t-1)} := \Omega^{(t-1)} - \frac{1}{K} \mathbf{1} \mathbf{1}^\top. \tag{48}$$

Note that $\mathbf{p}^\top \mathbf{r}^{(t-1)} = 0$ by definition of $\bar{f}^{(t-1)}$. Applying Cauchy–Schwarz we have:

$$\left| \mathbf{p}^\top \mathbf{V}^{(t-1)} \mathbf{r}^{(t-1)} \right| \le \left\| \mathbf{p}^\top \mathbf{V}^{(t-1)} \right\|_2 \left\| \mathbf{r}^{(t-1)} \right\|_2. \tag{49}$$

Due to the assumption that $\|\mathbf{V}^{(t-1)}\|_2 \le \rho$ and $\|\mathbf{p}\|_2 \le 1$ which is a probability vector, we have:

$$\left| \mathbf{p}^\top \mathbf{V}^{(t-1)} \mathbf{r}^{(t-1)} \right| \le \rho \left\| \mathbf{r}^{(t-1)} \right\|_2. \tag{50}$$

Next bound $\|\mathbf{r}^{(t-1)}\|_2$ by the mean gap as:

$$\begin{aligned}
\left\| \mathbf{r}^{(t-1)} \right\|_2^2 &= \sum_k \left( f_k^{(t-1)} - \bar{f}^{(t-1)} \right)^2 \\
&\le \sum_k \left( f_k^{(t-1)} \right)^2 \\
&\le \left( \max_k f_k^{(t-1)} \right) \sum_k f_k^{(t-1)} \\
&= \frac{\max_k f_k^{(t-1)}}{\min_k p_k} \left( \mathbf{p}^\top \mathbf{f}^{(t-1)} \right) \\
&\le \frac{1}{\min_k p_k}.
\end{aligned} \tag{51}$$

Let $c := 1/\sqrt{\min_k p_k} \le \sqrt{K}$, we can combine Eq. (50) and Eq. (51) to get the following inequality:

$$\left| \mathbf{p}^\top \mathbf{V}^{(t-1)} \mathbf{r}^{(t-1)} \right| \le \rho c \sqrt{\bar{f}^{(t-1)}}. \tag{52}$$

Then by squaring both sides of Eq. (52) and use $\sqrt{\bar{f}} \le 1 + \bar{f}$, we have:

$$\left| \mathbf{p}^\top \mathbf{V}^{(t-1)} \mathbf{r}^{(t-1)} \right| \le \rho^2 c^2 \left( 1 + \bar{f}^{(t-1)} \right) \le \frac{2\rho^2 c^2}{1 - \rho} \bar{f}^{(t-1)}, \tag{53}$$

where the last inequality employs $\bar{f}^{(t-1)} \le (1 - \rho)^{-1} \bar{f}^{(t-1)}$ which is trivial for $0 < \rho < 1$. Taking expectations, we can substitute Eq. (53) in to Eq. (48) to obtain:

$$\mathbb{E}\left[ \mathcal{L}\left(\Psi^{(t)}\right) - \mathcal{L}^\star \right] \le \left( 1 + \frac{2\rho^2}{1 - \rho} \right) \mathbb{E}\left[ \bar{f}^{(t-1)} \right]. \tag{54}$$

Given Eq. (43) and $\bar{f}^{(t-1)} = \sum_k p_k \mathbb{E}\left[ f_k^{(t-1)} \right]$, we can bound $\mathbb{E}[f_k^{(t-1)}]$ by:

$$\mathbb{E}\left[ f_k^{(t-1)} \right] \le (1 - \eta\mu)^Q \left( \mathcal{L}_k\left(\Psi_k^{(t-1)}\right) - \mathcal{L}_k^\star \right) + \frac{\eta \mathscr{L} \zeta^2}{2\mu}. \tag{55}$$

By taking a weighted sum of the above formula over all $K$ clients with weights $p_k$, we obtain:

$$\mathbb{E}\left[ \bar{f}^{(t-1)} \right] \le (1 - \eta\mu)^Q \left( \mathcal{L}\left(\Psi^{(t-1)}\right) - \mathcal{L}^\star \right) + \frac{\eta \mathscr{L} \zeta^2}{2\mu}. \tag{56}$$

**One-round contraction**    By plugging Eq. (56) into Eq. (54), we have:

$$\mathbb{E}\left[\mathcal{L}\left(\Psi^{(t)}\right) - \mathcal{L}^{\star}\right] \le (1 - \eta\mu)^Q \left(1 + \frac{2\rho^2}{1-\rho}\right)\left(\mathcal{L}\left(\Psi^{(t-1)}\right) - \mathcal{L}^{\star}\right) + \frac{\eta\mathscr{L}\zeta^2}{2\mu}\left(1 + \frac{2\rho^2}{1-\rho}\right). \tag{57}$$

Since $1 + \frac{2\rho^2}{1-\rho} \le 1 + \frac{2\rho}{1-\rho} = \frac{1}{1-\rho}$ and $\rho < 1$, the following inequality holds:

$$\mathbb{E}\left[\mathcal{L}\left(\Psi^{(t)}\right) - \mathcal{L}^{\star}\right] \le (1 - \eta\mu)^Q \left(\mathcal{L}\left(\Psi^{(t-1)}\right) - \mathcal{L}^{\star}\right) + \frac{\eta\mathscr{L}\zeta^2}{2\mu} + \frac{2\eta\mathscr{L}\rho^2}{\mu(1-\rho)^2}, \tag{58}$$

where the last term absorbs the factor from $(1-\rho)^{-1}$.

**Across $T$ communication rounds**    By setting $\gamma := (1 - \eta\mu)^Q$ where $0 < \gamma < 1$, we can unroll Eq. (58) as:

$$\mathbb{E}\left[\mathcal{L}\left(\Psi^{(T)}\right) - \mathcal{L}^{\star}\right] \le \gamma^T \left(\mathcal{L}\left(\Psi^{(0)}\right) - \mathcal{L}^{\star}\right) + \frac{\eta\mathscr{L}\zeta^2}{2\mu}\sum_{t=0}^{T-1}\gamma^t + \frac{2\eta\mathscr{L}\rho^2}{\mu(1-\rho)^2}\sum_{t=0}^{T-1}\gamma^t. \tag{59}$$

Since the geometric sums satisfy $\sum_{t=0}^{T-1}\gamma^t \le \frac{1}{1-\gamma}$, while $1 - \gamma = 1 - (1 - \eta\mu)^Q \ge \eta\mu$, we have:

$$\sum_{t=0}^{T-1}\gamma^t \le \frac{1}{\eta\mu}. \tag{60}$$

By inserting the above inequality into Eq. (59) and simplify, we can easily get:

$$\mathbb{E}\left[\mathcal{L}\left(\Psi^{(T)}\right) - \mathcal{L}^{\star}\right] \le \gamma^T \left(\mathcal{L}\left(\Psi^{(0)}\right) - \mathcal{L}^{\star}\right) + \frac{\eta\mathscr{L}\zeta^2}{2\mu} + \frac{2\eta\mathscr{L}\rho^2}{\mu(1-\rho)^2}. \tag{61}$$

Recovering $\gamma^T = (1 - \eta\mu)^{QT}$ gives exactly Eq. (12) in Theorem 3.3, which concludes the proof.

# D    Experimental Details

## D.1    Dataset Statistics

Table 4 summarizes the statistics of the datasets used in our experiments. It includes the total number of nodes, edges, node classes, and feature dimensions for each dataset. Specifically, we use four citation graph datasets (Cora, CiteSeer, Pubmed, and ogbn-arxiv) and two product co-purchase graph datasets (Amazon-Computer and Amazon-Photo).

Table 4: Dataset statistics

| Datasets | Nodes | Edges | Classes | Features |
|---|---|---|---|---|
| Cora | 2,708 | 5,429 | 7 | 1,433 |
| CiteSeer | 3,327 | 4,732 | 6 | 3,703 |
| Pubmed | 19,717 | 44,324 | 3 | 500 |
| Amazon-Computer | 13,752 | 491,722 | 10 | 767 |
| Amazon-Photo | 7,650 | 238,162 | 8 | 745 |
| ogbn-arxiv | 169,343 | 2,315,598 | 40 | 128 |

## D.2    Implementation Details

Following the standard FL settings, in our `FedAux`, both the local client models and the global server model adopt the same backbone architecture. We employ MaskedGCN [3] to generate node embeddings and sweep the number of GCN layers over $L \in \{1, 2, 3\}$. The hidden dimension is selected from $d' \in \{64, 128, 256\}$, and dropout probabilities are set to 0.5. The auxiliary projection vector $\boldsymbol{a}$ is initialized from a Gaussian distribution in $\mathbb{R}^{d'}$. The similarity–temperature parameter $\alpha$ is set to 10, and the bandwidth $\sigma$ is fixed to 1. For the FL schedule, we run $T = 100$ communication rounds with $Q = 1$ local epoch on the smaller citation datasets (Cora, CiteSeer, Pubmed). On all other datasets, we set the total number of rounds to $T = 200$ and the number of local epochs per round to $Q = 2$. All experiments are executed on a workstation equipped with an NVIDIA Tesla V100 SXM2 GPU (32 GB) running CUDA 12.4.

Table 5: The degree of non-IIDness.

| Non-IIDness | Cora | | | CiteSeer | | | Pubmed | | |
|---|---|---|---|---|---|---|---|---|---|
| | 5 Clients | 10 Clients | 20 Clients | 5 Clients | 10 Clients | 20 Clients | 5 Clients | 10 Clients | 20 Clients |
| $\xi$ | 0.2667 | 0.3092 | 0.3760 | 0.1848 | 0.2292 | 0.2572 | 0.1316 | 0.1500 | 0.1725 |

| Non-IIDness | Amazon-Computer | | | Amazon-Photo | | | ogbn-arxiv | | |
|---|---|---|---|---|---|---|---|---|---|
| | 5 Clients | 10 Clients | 20 Clients | 5 Clients | 10 Clients | 20 Clients | 5 Clients | 10 Clients | 20 Clients |
| $\xi$ | 0.2774 | 0.3582 | 0.3931 | 0.3600 | 0.4314 | 0.4840 | 0.3398 | 0.3668 | 0.4307 |

## D.3 Quantifying Non-IIDness in Federated Graph Datasets

To compare how much statistical heterogeneity (i.e., non-IIDness) each dataset induces under a given partition scheme, it is useful to measure how far the local data distribution at each client deviates from the global distribution and how dispersed local distributions are from one another. Below are three complementary, fully formalized metrics that can be computed once the graph has been split into $K$ client subgraphs $\{G_1, \cdots, G_K\}$.

**Label-distribution divergence**   Let $P(y)$ be the global class prior and $P_k(y)$ the class prior in client $G_k$. We use the average Jensen–Shannon (JS) divergence to measure the gap between each local label prior and the global label prior as follows:

$$\text{JSD} = \frac{1}{K} \sum_{k=1}^{K} \frac{1}{2} \left[ \text{KL} \left( P_k \| R_k \right) + \text{KL} \left( P \| R_k \right) \right], \tag{62}$$

where $R_k = \frac{1}{2}(P_k + P)$ denotes the mid-point distribution between $P_k$ and $P$ with $R_k(y \in Y) = \frac{1}{2} \left[ P_k(y) + P(y) \right]$. Pinsker's inequality [6] gives that $\text{JSD} \in [0, \log 2]$. A small JSD indicates that each client's label distribution closely matches the global prior, so the partition is effectively IID. As the JSD increases, local class proportions deviate more sharply from the global mixture, making individual clients progressively class-specific and therefore increasingly subject to statistical non-IIDness.

**Subgraph-distribution discrepancy**   Label skew alone may underestimate heterogeneity when covariate shift is strong. To quantify covariate-shift–induced heterogeneity, we measure how far the embedding distribution of each client deviates from that of every other client in an embedding space that reflects graph structure and attributes. We first obtain node embeddings for all nodes with a simple neighbor aggregation $\mathbf{Z}_k = \mathbf{A}_k \mathbf{X}_k = \{z_{k,i}\}_{i=1}^{N_k}$ in each client. Let $\mathbf{Z}_k$ be the empirical distribution of embeddings held by client $G_k$. We define the graph-distribution discrepancy of a $K$-client partition as the mean pair-wise maximum mean discrepancy (MMD) as:

$$\text{MMD} = \frac{2}{K(K-1)} \sum_{1 \leq k < l \leq K} \left\| \frac{1}{|V_k|} \sum_{v_i \in V_k} \phi(z_{k,i}) - \frac{1}{|V_l|} \sum_{v_j \in V_l} \phi(z_{l,j}) \right\|_2^2, \tag{63}$$

where $\phi(\cdot)$ is the canonical feature map of a Gaussian RBF kernel with the bandwidth fixed with the median pairwise distance to ensure comparability across datasets. Eq. (63) evaluates to zero when all clients share an identical embedding distribution (IID) and increases monotonically with covariate divergence.

Based on the above two perspectives, we can quantify the degree of non-IIDness by summing JSD and MMD as $\xi = \text{JSD} + \text{MMD}$, where a higher $\xi$ indicates a higher degree of non-IIDness. We show the degree of non-IIDness of all datasets under different numbers of clients in Table 5.

## D.4 Synthetic Graph for Client Similarity Estimation

We first generate an SBM graph with 3000 nodes that are uniformly divided into $K = 20$ equal-sized blocks as clients. Inter-client edges are added with probability $P^{\text{inter}} = 0.02$. To inject structural non-IIDness, the 20 clients are grouped into five super-clusters $\{\mathcal{G}_i\}_{i=1}^5$ (four clients per group). For every client belonging to $\mathcal{G}_i$, we draw intra-client edges with probability $P_i^{\text{intra}} = 0.15 \times i$, so clients

Table 6: Ablation studies on the federated node classification task under 10 clients.

| Baseline | Cora | Pubmed | Amazon-Computer | ogbn-arxiv |
|---|---|---|---|---|
| (i) FedAux$_{\text{hard}}$ | $80.29_{\pm 0.71}$ | $83.06_{\pm 0.35}$ | $88.10_{\pm 1.02}$ | $66.53_{\pm 0.12}$ |
| (ii) FedAvg$_{\text{mask}}$ | $78.98_{\pm 0.54}$ | $83.97_{\pm 0.51}$ | $84.31_{\pm 0.60}$ | $65.09_{\pm 0.07}$ |
| FedAux | $\mathbf{82.05_{\pm 0.71}}$ | $\mathbf{85.43_{\pm 0.29}}$ | $\mathbf{89.92_{\pm 0.15}}$ | $\mathbf{68.50_{\pm 0.27}}$ |

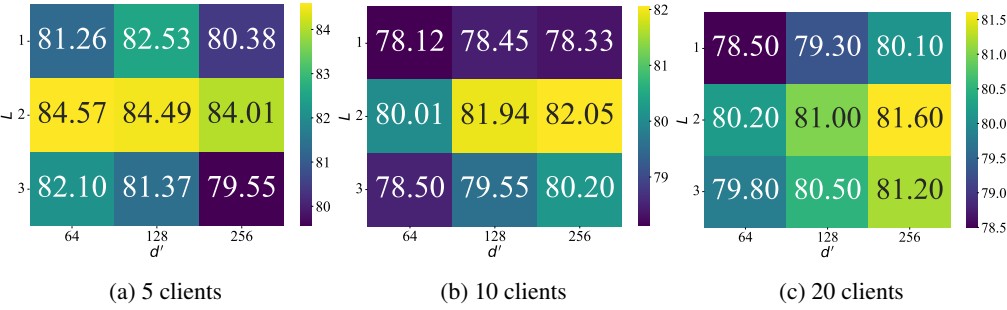

(a) 5 clients      (b) 10 clients      (c) 20 clients

Figure 7: Classification accuracy (%) for different GCN configurations.

within the same group are structurally IID, while clients across groups are non-IID. To inject label and feature non-IIDness, we assign 5 labeled classes to all nodes. Nodes owned by the group $\mathcal{G}_i$ receive label $i-1$ with high probability 0.8, and the remaining 0.2 mass is distributed uniformly over the other four labels. Each node's feature vector is the one-hot encoding of its label. In this way, clients in the same group are IID in both label and feature space, while clients from different groups exhibit pronounced distributional shifts. This controlled setting allows us to test whether the learned APVs can cluster clients that are genuinely similar. We directly compute the similarity between clients' mean-pooling embeddings without considering privacy in Fig. 3a as the ground-truth similarity. Fig. 3b-3d are similarities computed by APVs, learnable weights of the readout layer, and functional embedding [3].

# E  More Experiments

## E.1  Ablation Study

We conduct an ablation study in Table 6 validate our motivation and design, with the following two ablations: (i) replacing our proposed continuous aggregation scheme over the $\boldsymbol{a}_k$-space defined in Eq. (6) with the $\mathrm{Conv1D}$ operation applied on the hard-sorted embeddings as introduced by [18], yielding a variant FedAux$_{\text{hard}}$, and (ii) removing our server-side APV-based personalized aggregation, instead using simple averaging to aggregate local models into a global model, leading to the baseline FedAvg$_{\text{mask}}$. Note that the FedAvg$_{\text{mask}}$ variant differs from the standard FedAvg used in Table 1, where FedAvg$_{\text{mask}}$ adopts MaskedGCN [3] as the GNN backbone, whereas the standard FedAvg utilizes a conventional GCN architecture. Compared with FedAux$_{\text{hard}}$, our FedAux consistently obtains higher accuracy. This empirical gain confirms our claim that although the hard-sorting scheme proposed by [18] does allow gradient flow to optimize the APV, the underlying discrete permutation remains non-differentiable and therefore restricts the capacity of the APV to adapt. By using our proposed continuous aggregation with a fully differentiable kernel operator, FedAux enables smoother gradient flow, allowing the APV and the local GNN to co-evolve optimally. Besides, FedAux and FedAvg$_{\text{mask}}$ both adopt MaskedGCN as backbone, while FedAux personalized aggregates local models based on the APV similarity, while FedAvg$_{\text{mask}}$ aggregates local models to a single global model. Results show that FedAux outperforms FedAvg$_{\text{mask}}$ on all datasets, which demonstrates that exploiting APV-based personalized aggregation allows the federation to respect non-IID data distributions and learn more effective client-specific models.

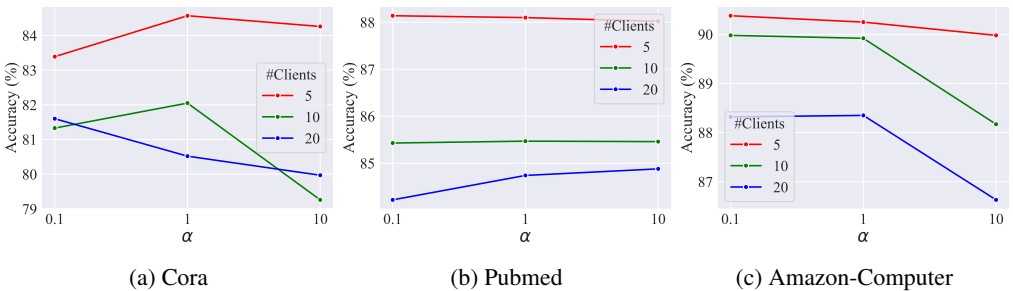

Figure 8: Sensitivity of `FedAux` on the similarity temperature parameter $\alpha$.

## E.2 Hyperparameter Analysis

**Impact of Model Depth and Hidden Dimension**  In our model, the number of GCN layers is selected from $L \in \{1, 2, 3\}$, and the dimension of the hidden layers is selected from $d' \in \{64, 128, 256\}$. In Fig. 7, we show all the hyperparameter combinations on the Cora dataset for different client counts. It is evident that `FedAux` consistently achieves the highest performance with $L = 2$ across all federated settings. For the hidden dimension, when the number of clients is not large, `FedAux` requires a relatively small hidden dimension, while with 20 clients, a hidden dimension of 256 yields the best results.

**Impact of Similarity Temperature** $\alpha$  In Eq. (7), the similarity temperature parameter $\alpha$ is introduced to modulate the sharpness of the similarity distribution. To evaluate the sensitivity of `FedAux` to this hyperparameter, we test $\alpha \in \{0.1, 1, 10\}$ across varying numbers of clients and report the resulting accuracy in Fig. 8. The results show that while some configurations achieve optimal accuracy at different values of $\alpha$, for example, $\alpha = 0.1$ is optimal for 20 clients on Cora and $\alpha = 10$ is optimal for 20 clients on Pubmed, directly setting $\alpha = 1$ consistently provides satisfactory performance across different datasets and client counts.

## F  Limitations and Future Work

The main limitation of our work is that while the kernel-based continuous aggregation scheme lets each client learn a fully differentiable, data-driven ordering on the `APV`, it requires evaluating the pairwise kernel for every node pair in the local client subgraph. Thus, the per-client computational cost scales quadratically, $\mathcal{O}(N_k^2)$, which is acceptable for small and medium subgraphs but can dominate run-time on very large clients. Our future work focuses on addressing this scalability bottleneck. One promising direction is to approximate the dense kernel with low-rank factorizations, such as Nyström approximation, so that complexity scales linearly in $N_k$ with controllable error. We stress that our core contribution is a privacy-preserving mechanism for personalized clustering clients via the `APV`, and reducing local computational load is a complementary line of future research.

