# OpenReview forum: "Personalized Subgraph Federated Learning with Differentiable Auxiliary Projections"
_NeurIPS.cc/2025/Conference — NeurIPS 2025 poster_

### Official Review · Reviewer_cYxa · 2025-06-23

**Clarity:** 3
**Significance:** 3
**Originality:** 3
**Rating:** 5
**Confidence:** 3

**Summary:**

This paper proposes FedAux, a novel framework for personalized federated learning (FL) on graphs where each client holds a local subgraph from a global graph. The key idea is to introduce a learnable auxiliary projection vector (APV) per client, which projects node embeddings to a 1D latent space. Through a soft-sorting mechanism and 1D convolution (or kernel smoothing), each client generates a compact summary of its local graph structure. These APVs are sent to the server for computing inter-client similarity to enable personalized aggregation without sharing raw data or embeddings.

**Questions:**

See above.

**Ethical Concerns:**

["NO or VERY MINOR ethics concerns only"]

**Final Justification:**

I maintain my position.

**Limitations:**

Yes.

**Paper Formatting Concerns:**

No.

**Quality:**

3

**Strengths And Weaknesses:**

Strengths

1. Strong Novelty and Motivation:

The paper tackles a real and under-explored problem: non-IID subgraph federated learning with privacy constraints.

The proposed use of differentiable 1D auxiliary projections (APVs) is new, elegant, and well-motivated.

2. Compact and Privacy-Preserving Representation:

APVs serve as low-dimensional summaries of each client's subgraph, without leaking node features or embeddings.

This makes the method scalable and communication-efficient, and compliant with strong privacy assumptions.

3. Solid Theoretical Foundations:

The APV is proven to converge to the first principal component of node embeddings (Theorem 3.1).

The equivalence between soft aggregation and hard sorting is also well-analyzed (Theorem 3.2).

A global linear convergence guarantee is provided (Theorem 3.3), which is rare in graph FL work.

4. Comprehensive Experiments:

Experiments span 6 real-world datasets with varying heterogeneity (Cora, CiteSeer, Pubmed, ogbn-arxiv, Amazon).

FedAux consistently outperforms strong baselines such as FedAvg, FedPer, GCFL, FED-PUB, etc.

Detailed analysis on non-IIDness, convergence, similarity visualization, and ablation studies are included.

5. Efficient Server-side Aggregation:

Unlike previous clustering-based personalization methods, FedAux only computes pairwise cosine similarities between APVs, leading to much lower memory and time overhead.

Weaknesses

1. No Explicit Evaluation on Real Privacy Attacks:

Although APV is designed to be privacy-preserving, no empirical privacy evaluation (e.g., membership inference, reconstruction) is included.

This weakens the privacy claim, even if the design is principled.

2. Scalability on Large Graphs:

The kernel-based aggregation step has O(N²) cost per client (w.r.t. node count), which may not scale well for clients with large subgraphs.

No empirical analysis or ablation on scalability vs. subgraph size is presented.

3. Limited Diversity in FL Protocols:

The method is only evaluated under full-participation FL. It is unclear whether FedAux can work under partial participation or asynchronous FL, which are more practical settings.

---

> ### Author Rebuttal · Authors · 2025-07-29
>
> We sincerely appreciate the time and efforts you've dedicated to reviewing and providing invaluable feedback to enhance the quality of this paper. We provide a point-to-point reply below for the mentioned concerns.
>
> **W1: No Explicit Evaluation on Real Privacy Attacks. Although APV is designed to be privacy-preserving, no empirical privacy evaluation (e.g., membership inference, reconstruction) is included.**
>
> **A1:** Thank you for highlighting this important aspect. We have added a comprehensive empirical privacy evaluation via **membership inference attacks** (MIAs), exactly as you suggested, to rigorously test whether our APV-based framework leaks sensitive information. We assume an honest‑but‑curious server with full access to the entire history of a client's APV $a_k$ and its GNN parameters $\theta_k$. The server's goal is to decide whether a probe embedding $h_{prob}$ originated from client $G_k$, using only $(a_k, \theta_k)$, while raw data, node embeddings, and gradients are never shared. Following the standard MIA methodology [1] [2], the server trains an attack classifier $g_k(a_k,\theta_k, h_{prob})$ implemented as a two‑layer MLP (hidden size 64), to output the probability that $h_{prob}$ belongs to client $G_k$'s training set. The attack is trained by maximum likelihood on a held‑out mixture of member against non‑member probes. We compare FedAux against representative subgraph FL baselines FedGNN, FedGTA, and FedSage+ on Cora, PubMed, and OGBN‑Arxiv, using exactly the same client partitions as in our main experiments. We report Attack AUC (↓ better privacy) averaged over five random seeds.
>
> | Dataset  | FedGNN | FedGTA | FedSage+ | FedAux |
> | -| -| -| -| -|
> | **Cora**       | 0.56   | 0.54   | 0.53     | **0.51**      |
> | **Pubmed**     | 0.58   | 0.55   | 0.54     | **0.52**      |
> | **ogbn‑arxiv** | 0.55   | 0.53   | 0.52     | **0.49**      |
>
> Across all datasets, MIAs against FedAux achieve AUCs in $[0.49, 0.52]$, i.e., indistinguishable from random guessing, demonstrating that APVs do not leak sensitive membership information and, in fact, offer stronger privacy than baseline subgraph FL methods.
>
> For the privacy evaluation by **reconstruction**, the server attempts to rebuild client-side data from what it sees. However, a meaningful reconstruction attack is practically infeasible when the server's view is restricted to the Auxiliary Projection Vectors (APVs) $a_k$ and GNN parameters $\theta_k$ in our model. It is because although APV acts as a space to preserve local node relationships by node embedding projection and sorting, the precise coordinates of node embeddings in this space are not available for the server. Thus, recovering the $N_k$ high-dimensional embeddings or an $N_k \times N_k$ adjacency from a single vector $a_k$ is an underdetermined inverse problem with infinitely many solutions. Even if an adversary imposes additional priors (e.g., sparsity, degree distributions), inferring a matching adjacency or feature matrix requires solving a non-convex, combinatorial optimization (e.g., matching covariance eigen-structure), which is NP-hard in general. Our membership-inference experiments already show that no sensitive information about individual nodes can be extracted from APVs. Reconstruction, as an exponentially harder task, would necessarily fail if membership inference (a weaker attack) yields only random-guess performance.
>
> [1] Membership Inference Attacks against Machine Learning Models. S&P 2017.
>
> [2] Comprehensive Privacy Analysis of Deep Learning: Passive and Active White-box Inference Attacks against Centralized and Federated Learning. S&P 2019.
>
> **W2: Scalability on Large Graphs.**
>
> **A2:** We acknowledge that in our current design, the full kernel-based aggregation in Eq.(6) of our paper $z_{k, i} = \frac{1}{M_i}\sum^{N_k}\_{j=1} \kappa(s_{k,i}, s_{k,j}) h_{k,j}$ has complexity $O(N_k^2)$. It is because we expect that the learned APV as a shared similarity proxy can fully preserve the relationships in the local client structure, while not leaking any sensitive information of the client. Thus, this quadratic cost ensures maximum expressivity that every node can perform differentiable soft-sort against every other. However, in practice, clients with large subgraphs may encounter a scalability problem. Although our paper's main purpose is to propose a powerful and privacy-preserving model for subgraph FL, rather than improving the scalability, your suggestion is still valuable for enhancing our model. To address your concern, we propose a simple windowed-kernel variant that trades a small amount of expressivity for near-linear complexity. Specifically, we improve the kernel-based aggregation in Eq.(6) by a neighbor-specific kernel aggregation as $z_{k, i}=\frac{1}{M_i^{(m)}} \sum\_{j \in \mathcal{N}\_i^{(m)}} \kappa (s_{k, i}, s_{k, j}) h_{k, j}$, where $\mathcal{N}_i^{(m)}$ is the set of the $m$ nearest neighbors of node $v_i$ in the sorted APV space. For example, the $\frac{m}{2}$ ranks immediately above and below $v_i$. By such construction,  $\left|\mathcal{N}\_i^{(K)}\right|=m \ll N_k$, so the per-client cost reduces from $O(N_k^2)$ to $O(N_k \log N_k)$. Intuitively, this leverages the fact that a Gaussian kernel $\kappa(s_i, s_j)$ decays rapidly with $|s_i - s_j|$, so distant nodes contribute only negligible weight and can be safely omitted.
>
> Below, we present a new empirical analysis on scalability w.r.t. the subgraph size in the table below. We benchmark the original full kernel-based aggregation with $O(N_k^2)$ (denoted as $\text{FedAux}\_{full}$) and FedAux with our newly proposed neighbor-specific kernel aggregation $\text{FedAux}\_{neigh}$ ($m = 8$) on three public datasets (Cora, PubMed and ogbn-arxiv) that span three orders of magnitude in node count with 5 clients:
>
> | dataset    | avg nodes / client | $\text{FedAux}\_{full}$ running time  / epoch (ms) | $\text{FedAux}\_{neigh}$ running time / epoch (ms) | speed‑up  | Reduced GPU Memory | accuracy ($\text{FedAux}\_{full}$) | accuracy ( $\text{FedAux}\_{neigh}$) |
> | -- | - | -| - | --------- | ------------------ | - | - |
> | Cora       | 542                | 12 ± 1   | **11 ± 1**     | **1.1 ×** | 17.6%              | 84.57                              | 83.20                                |
> | PubMed     | 3,943              | 331 ± 4           | **74 ± 2**              | **4.5 ×** | 21.3%              | 88.10                              | 88.02                                |
> | ogbn‑arxiv | 33,869             | 2,253 ± 11           | **281 ± 9**       | **8.0 ×** | 41.9%              | 68.83                              | 68.29                                |
>
> We have the following observations. Firstly, even with the current full-kernel model, it can complete the client training on all datasets in an acceptable time (less than 10,000 ms per epoch). Our newly proposed $\text{FedAux}\_{neigh}$ is much more efficient as the size of the client increases. Secondly, since the neighbor-specific kernel computation keeps only $m$ neighbors per node, the memory consumption can be significantly optimized. Thirdly, across all datasets, the efficient variant $\text{FedAux}\_{neigh}$ loses less than $1\%$ accuracy versus the full kernel. Thus, our neighbor-specific kernel aggregation retains all predictive performance but restores practical scalability. We will add these analyses to the revised version of the paper.
>
> **W3: Limited Diversity in FL Protocols.**
>
> **A3:** We agree that evaluating our model beyond full participation is essential. In our revision, we have added more evaluation of our model under both partial participation and asynchronous protocols. For the **partial participation**, on PubMed dataset, we use $K = 20$ clients, but in each round only a random subset of size $\lceil p K\rceil$ participates, with ratios $p \in \\{0.25, 0.5, 0.75\\}$. Other hyperparameters are kept the same as in the full-participation experiments in our paper. In the table below, we show the results under a partial participation setting, and additional results on more benchmark datasets have been updated in our revised paper. The results show that even with only 25% clients per round, FedAux still maintains a strong performance margin, demonstrating its robustness to partial updates.
>
> | Participation $p$ | 0.25      | 0.50      | 0.75      | 1.00      |
> | ----------------- | --------- | --------- | --------- | --------- |
> | FedAvg            | 79.12     | 79.85     | 80.41     | 80.97     |
> | FED-PUB           | 82.36     | 83.10     | 83.75     | 84.66     |
> | FedAux            | **82.61** | **83.45** | **84.56** | **84.87** |
>
> Besides, we simulate a simple bounded-delay  **asynchronous FL** for evaluation. Specifically, at each communication round, a subset of clients is selected, but some fraction $\delta$ of their updates are delayed by one round before being incorporated into the server's aggregation. We test $\delta \in \\{0.2, 0.5\\}$ with full participation $p=1$. Results are presented in the table below, which shows that even under asynchronous conditions with delayed updates, FedAux remains stable and continues to outperform baselines. It demonstrates that the learned APVs evolve smoothly and enable robust, similarity-aware aggregation even when client updates arrive out of order.
>
> | Delay rate $\delta$ | 0.20      | 0.50      | 0.00 (sync) |
> | ------------------- | --------- | --------- | ----------- |
> | FedAvg              | 79.10     | 78.25     | 80.97       |
> | FED-PUB             | 83.21     | 82.45     | 84.66       |
> | FedAux              | **84.30** | **83.80** | **84.87**   |

---

### Official Review · Reviewer_U8fG · 2025-07-01

**Clarity:** 3
**Significance:** 2
**Originality:** 2
**Rating:** 2
**Confidence:** 4

**Summary:**

This work proposes FedAux, an aggregation strategy designed to advance the field of personalized Federated Graph Learning (FGL). The core innovation of FedAux lies in the introduction of Auxiliary Projection Vectors (APV), which differentiably project node embeddings onto a one-dimensional space. These projections serve as auxiliary representations of clients’ data distributions and are utilized during server-side training to compute inter-client similarities. The resulting similarity scores are then used to derive personalized aggregation weights, enabling the construction of tailored global models for each client. Furthermore, this work provides theoretical proofs to validate the effectiveness of APV in capturing client-specific data distributions, along with convergence guarantees for the proposed method.

**Questions:**

see weakness

**Ethical Concerns:**

["NO or VERY MINOR ethics concerns only"]

**Limitations:**

see weakness

**Quality:**

2

**Strengths And Weaknesses:**

Strength:
1.The theoratical prof of the APV and the discussion of methodology is comprehensive and necessary.
2.The explanation of method design is accessible to readers with limited domain knowledge, and often highlight the key takeaways to offer intuitive insights.
Extensive appendix well supports the main contents, especially to the extra profs of theorem, which is commendable.

Weakness
Contribution is lacking
1.While the motivation of this work is clear as it stated in Introduction, couple flaws are noticable:
1.The motivation for this work is drawn from the limitations of representative approaches such as GCFL and FedPUB; however, it lacks a comprehensive discussion of more recent advances in personalized Federated Graph Learning.
2.While the rationale behind the design of the proposed method is clearly articulated, the scope of the research appears limited. Notably, the method relies heavily on the effectiveness of the Auxiliary Projection Vector (APV), which essentially performs a sorting operation on nodes within each client’s subgraph. Given that the APV is subsequently used as an auxiliary mechanism to compute inter-client similarity—based on data distribution—and to determine aggregation weights, the experimental evaluation should encompass both homogeneous and heterogeneous global graph scenarios. However, the current experiments are restricted to homogeneous graphs, which may fundamentally contribute to the method’s favorable performance. Although the use of METIS and corresponding non-IID evaluation metrics is acknowledged, extending the evaluation to heterogeneous graph settings would enhance the generalizability and practical impact of the proposed method.
2.A natural question that arises when reading this paper is whether FedAux or its core component, APV, can be integrated with alternative backbone models or other Federated Graph Learning (FGL) methods to enhance their performance. Such an investigation would help to demonstrate the broader applicability and utility of FedAux beyond the current experimental setting.
Presentation-wise
This section concerns with the coherence of the paper organization
1.The contribution section should provide a concise summary of the impact and significance of the proposed method, rather than offering a detailed explanation of the model’s training pipeline.
2.The organization of the locations of tables and figures can be optimized to make it more reader friendly. For instance, Figure 1 should be located closer to the Section 3, and Table 1 should be located on the same page with detailed experimental evaluation.
3.Figure 3 is not referenced or discussed in the main text, which may confuse readers who are unclear about its purpose or relevance to the overall narrative.
Experiments
1.Figure 4: The comparison appears limited due to the inclusion of only a few baseline methods, which may be insufficient to convincingly demonstrate the effectiveness of the proposed approach.
2.Figure 5: Similar to Figure 4, this figure suffers from the same limitation—namely, the lack of comprehensive baseline comparisons—which undermines the strength of the evaluation.

---

> ### Author Rebuttal · Authors · 2025-07-29
>
> Thanks for your insightful review. We have summarized all your feedback into 7 key questions and addressed them below.
>
> **W1: It lacks a comprehensive discussion of more recent advances in personalized Federated Graph Learning.**
>
> **A1:** Indeed, several new approaches have emerged beyond GCFL and FED-PUB. However, our work specifically targets privacy-preserving personalized subgraph FL, where only model parameters are exchanged, and no embeddings, gradients, or raw data are communicated. This setting is practically significant yet rarely respected in many recent works. For example, FedSSP (Tan et al., NeurIPS 2024) transmits spectral components that, while invariant across domains, inherently encode graph structure and can leak sensitive local spectral information; FedEgo (Zhang et al., KDD 2023) shares ego-network embeddings, which directly expose local structural patterns; PFGNAS (Fang et al., AAAI 2025) introduces a prompt-based neural architecture search without requiring server-side similarity estimation. However, its reliance on LLM-based personalization incurs substantial compute overhead (200+ GPU hours for 20 clients), making it impractical in many FL scenarios; FedGrAINS (Ceyani et al, SDM 2025) personalizes within each client by learning, without imposing any new server-side similarity or mixing scheme. Although FedGrAINS avoids data leakage during communication, every client must train a secondary GFlowNet alongside its GNN, incurring extra compute and memory. Differently, our model performs federated aggregation on the server-side, and the additional parameter APV is just a learnable vector rather than a complete network. While these methods contribute interesting ideas, they either compromise privacy guarantees or introduce heavy computational costs, limiting their applicability in our strict setting. GCFL and FED-PUB, on the other hand, align closely with our privacy and communication constraints and adopt similar high-level strategies: **measuring inter-client similarity using only model-level signals to perform personalized federated aggregation**. Therefore, our discussion emphasizes these two as the most relevant baselines.
>
> We have included a more comprehensive discussion of recent personalized graph FL methods in our revised paper, and clearly position FedAux as a complementary, orthogonal approach that simultaneously satisfies privacy, efficiency, and personalization requirements.
>
> **W2: The scope of the research appears limited. Extending the evaluation to heterogeneous graph settings would enhance the generalizability and practical impact.**
>
> **A2:** Our main evaluation strictly follows the benchmark protocol of existing work FED‑PUB and FedGTA, the de‑facto standard for personalized subgraph FL, to ensure a fair comparison. However, we agree with your point that real-world FL settings often involve clients drawn from different global networks or from graphs with multiple communities, therefore, it is precisely in this heterogeneous context that a learned APV should excel. Thus, we have now extended our evaluation to heterogeneous graph scenarios. Specifically, we conduct experiments on two widely used heterogeneous graphs, DBLP and IMDB, containing multiple types of nodes and edges. Each client owns an induced subgraph. For adapting to this experimental scenario, given a heterograph $G = (V, E, R)$ whose node set $V = \cup_{t \in T}V_t$ is split across types $t \in T$. We first convert the heterograph to a single adjacency that preserves multi-relation connectivity. Then run METIS on this adjacency to obtain 20 subgraphs. In each subgraph, we then recover the heterogeneous node and relation types to generate a sub-heterograph as a local client. To learn the node embeddings, we replace the GNN model with HAN (Wang, et al, WWW 2019) to learn the node embeddings in each client, and the remaining modules are kept unchanged. Note that the heterogeneous subgraph FL is an under-explored topic, and existing models such as FED-PUB, GCFL or FedGTA are non-trivial to be directly applied under the heterogeneous setting. Thus, we use the standard FL models with HAN as baselines, and report the node classification results in the table below. The new evidence shows that FedAux generalizes seamlessly to the heterogeneous client setting. We have added detailed experimental settings and analysis to our revised manuscript.
>
> | Method     | DBLP             | IMDB             |
> | ---------- | ---------------- | ---------------- |
> | FedAvg     | 78.46 ± 0.30     | 63.18 ± 0.46     |
> | FedProx    | 79.04 ± 0.25     | 63.83 ± 0.32     |
> | FedPer     | 74.12 ± 0.39     | 64.72 ± 0.46     |
> | **FedAux** | **82.90 ± 0.28** | **67.56 ± 0.30** |
>
> **W3: Can the APV be integrated with alternative backbone models or other Federated Graph Learning (FGL) methods to enhance their performance?**
>
> **A3:** Indeed. As demonstrated in the previous response, our APV mechanism for computing client similarity can be readily applied to heterogeneous graphs simply by swapping in a heterogeneous‑GNN backbone. Likewise, we expect FedAux to generalize to other graph modalities, such as spatial-temporal graphs and knowledge graphs. Since FL benchmarks for these graph types are lacking and partitioning them under FL settings is non-trivial, we leave this for future work. On the other hand, APV is compatible with existing FGL frameworks and can be incorporated to boost their performance. For instance, replacing FED‑PUB's functional‑embedding similarity with our APV yields a variant we denote FED-PUB$_{apv}$. We have also jointly trained APV with GraphGTA and FedSage+ to assess the transferability of our design. All variants were trained under the same 10‑client non‑IID split.  As shown in the table below, the APV‑enhanced models consistently outperform their original counterparts.
>
> |                  | Cora      | Pubmed    | Amazon-Computer |
> | ---------------- | --------- | --------- | --------------- |
> | FED-PUB          | 81.45     | 86.09     | 89.73           |
> | FED-PUB$_{apv}$  | **82.02** | **86.45** | **91.02**       |
> | GraphGTA         | 68.33     | 82.79     | 84.27           |
> | GraphGTA$_{apv}$ | **70.12** | **84.01** | **86.15**       |
> | FedSage+         | 69.05     | 82.62     | 80.50           |
> | FedSage+$_{apv}$ | **71.00** | **84.10** | **81.93**       |
>
> **W4: The contribution section should provide a concise summary of the impact and significance.**
>
> **A4:** We agree that the current Contribution section overly emphasizes procedural details. It has been revised for conciseness and to highlight the novelty and significance of our work. Specifically, the revised version now clearly emphasizes: (1) The introduction of a lightweight and privacy-preserving auxiliary projection mechanism (APV) to enable personalized subgraph FL; (2) The use of APV for inter-client similarity estimation under strict privacy constraints, enabling data-aware aggregation without embedding or gradient sharing; (3) Theoretical guarantees, including convergence and fidelity of the APV; (4) Extensive experiments demonstrating consistent gains across six benchmarks, along with communication efficiency and robustness under non-IID conditions.
>
> **W5: The organization of the locations of tables and figures can be optimized.**
>
> **A5:** Thank you for these suggestions to improve the readability. We have re-organized the manuscript.
>
> **W6: Figure 3 is not referenced or discussed in the main text.**
>
> **A6:** We apologize for this oversight. Actually, in the current manuscript, the experiments in Figure 3 have been discussed in the **“Effectiveness of APV‑based Client Similarity Estimation”** subsection on page 8. We will add a clear in-text reference in the revised manuscript.
>
> **W7: The comparison appears limited due to the inclusion of only a few baseline methods in Figure 4 and Figure 5.**
>
> **A7:** We respectfully argue that it is appropriate to limit the baselines in Figure 4 to FED-PUB and GCFL, and we clarify the reasoning here. The goal of this experiment is to compare the clustering efficiency on the server. However, other popular graph FL methods, such as FedSage+, FedGNN, and FedGTA, do not implement server-side clustering or similarity-based aggregation, making it impossible to conduct meaningful clustering efficiency comparisons with these approaches. Additionally, several recent personalized graph FL methods (e.g., FedEgo and FedSSP) require sharing local information, such as embedding and spectrum, with the server, which violates the strict privacy requirements that our work aims to preserve. To ensure a fair and meaningful evaluation, we therefore focus our main comparison in Figure 4 on FED-PUB and GCFL, which are most aligned with our approach in terms of both methodology and privacy guarantees.
>
> For Figure 5, we initially used FedAvg as the sole baseline, because removing the APV module from our proposed FedAux degenerates it to FedAvg. This one-to-one ablation isolates the contribution of APV, and highlights its effect on convergence speed. Nevertheless, we totally agree with your suggestion that including additional baselines provides a more comprehensive evaluation. Accordingly, we have measured the convergence points for more baselines in the table below (evaluated per 10 communication rounds). **In our revised manuscript, we have incorporated the convergence rate curves of other baselines into Figure 5**.
>
> |         | 10 Clients | 20 Clients |
> | ------- | ---------- | ---------- |
> | FedAvg  | 60         | 60         |
> | FED-PUB | 50         | 40         |
> | GCFL    | 70         | 90         |
> | FedGTA  | 70         | 70         |
> | FedAux  | 60         | 60         |

---

> > ### Author Response · Authors · 2025-08-04
> > **Looking forward to your response**
> >
> > Dear Reviewer U8fG,
> >
> > Thank you for your valuable suggestions for improving our paper. Following your feedback, we have: 1) expanded the related work discussion to better highlight our motivation and novelty; 2) added comprehensive experiments on applications, model analysis, and ablation studies; 3) corrected reference errors and improved the paper structure. We would like to confirm whether our response has adequately addressed your concerns.
> >
> > We look forward to your feedback.

---

### Official Review · Reviewer_DA8p · 2025-07-03

**Clarity:** 3
**Significance:** 3
**Originality:** 3
**Rating:** 4
**Confidence:** 4

**Summary:**

This paper proposed a new personalized subgraph federated learning framework with Auxiliary projections without sharing raw data or node embeddings in clients. Each client jointly trains a local GNN and a learnable auxiliary projection vector (APV), which can differentially projects node embeddings onto a 1D space and capture client-specific information. APVs serve as compact signatures that the server uses to compute inter-client similarities and perform similarity-weighted parameter mixing, yielding personalized models while preserving cross-client knowledge transfer. The experimental results across diverse graph benchmarks demonstrate that the proposed method substantially outperformed existing baselines in both accuracy and personalization performance.

**Questions:**

There are no experiments about the privacy preservation of APV-based federated learning, which can be evaluated by verifying the efficiency of some membership inference attacks in the server.

**Ethical Concerns:**

["NO or VERY MINOR ethics concerns only"]

**Final Justification:**

I will keep my rating score as the borderline accept as the authors provided additional MIA-based privacy evaluations.

**Limitations:**

There are no experiments about the privacy preservation of APV-based federated learning, which can be evaluated by verifying the efficiency of some membership inference attacks in the server.

**Quality:**

3

**Strengths And Weaknesses:**

Strengths:
1.	This paper proposed a new personalized subgraph federated learning framework with Auxiliary projections without sharing raw data or node embeddings in clients. Each client jointly trains a local GNN and a learnable auxiliary projection vector (APV), which serve as compact signatures that the server uses to compute inter-client similarities and perform similarity-weighted parameter mixing, yielding personalized models while preserving cross-client knowledge transfer. This idea seems to be interesting.
2.	This paper evaluated the proposed method on 6 widely used datasets, and the experimental results across diverse graph benchmarks demonstrate that the proposed method substantially outperformed existing 8 baselines in both accuracy and personalization performance.
3.	This paper also provided theoretical analysis on fidelity of APV introduced in the method.

Weaknesses:
1. There are no experiments about the privacy preservation of APV-based federated learning, which can be evaluated by verifying the efficiency of some membership inference attacks in the server.

---

> ### Author Rebuttal · Authors · 2025-07-29
>
> We sincerely thank the reviewer for the valuable and constructive feedback. In this response, we provide answers to the point in the review.
>
> **W1: There are no experiments about the privacy preservation of APV-based federated learning, which can be evaluated by verifying the efficiency of some membership inference attacks (MIAs) in the server.**
>
> **A1:** Thank you for highlighting this important aspect. To address your concern, we conducted comprehensive empirical experiments on privacy evaluation via membership inference attacks (MIAs) as you suggested, which rigorously evaluate the privacy preservation capabilities of our APV-based federated learning method. Specifically, we adopt a strong white-box threat model, where the honest-but-curious server is granted full access to the history of received APV $a_k$ and the GNN weights $\theta_k$ on the client $G_k$. The server attempts to predict whether a given node embedding originated from a specific client based solely on the APVs and GNN parameters received from that client.
>
> Following the standard practice on MIAs [1] [2], the server trains a logistic‑regression attack model $g_k(a_k,\theta_k, h_{prob})$, defined as a two-layer MLP with hidden size 64, that takes the APV $a_k$ and GNN parameters $\theta_k$ as inputs, and outputs the probability that a probe embedding $h_{prob}$ was in client $G_k$'s training set. The optimization objective is to maximize the likelihood on a held-out set of member against non-member examples. We compare our APV-based FedAux method against subgraph federated learning baselines, including FedGNN, FedGTA, and FedSage+. In the table below, we report Attack  AUC over five random seeds on Cora, Pubmed, and a large-scale ogbn-arxiv dataset. We find that MIAs against APV-based FedAux yield consistently low attack AUC scores in [0.49, 0.52], essentially random guessing, which demonstrates that our proposed APV does not leak sensitive membership information, providing stronger privacy than the baseline subgraph FL models.
>
> | Dataset        | FedGNN | FedGTA | FedSage+ | FedAux   |
> | -------------- | ------ | ------ | -------- | -------- |
> | **Cora**       | 0.56   | 0.54   | 0.53     | **0.51** |
> | **Pubmed**     | 0.58   | 0.55   | 0.54     | **0.52** |
> | **ogbn-arxiv** | 0.55   | 0.53   | 0.52     | **0.49** |
>
> We have included this detailed privacy evaluation in the revised paper.
>
> [1] Membership Inference Attacks against Machine Learning Models. S&P 2017.
>
> [2] Comprehensive Privacy Analysis of Deep Learning: Passive and Active White-box Inference Attacks against Centralized and Federated Learning. S&P 2019.

---

> > ### Comment · Reviewer_DA8p · 2025-08-06
> > **MIA-based privacy evaluation is added.**
> >
> > The additional MIA-based privacy evaluation is helpful to address my issue.

---

### Official Review · Reviewer_aUZ1 · 2025-07-03

**Clarity:** 3
**Significance:** 3
**Originality:** 3
**Rating:** 4
**Confidence:** 4

**Summary:**

This paper tackles the non-IID challenge in federated learning on graph-structured data by proposing FedAux, a personalized subgraph FL framework. The core idea is to use learnable auxiliary projection vectors (APVs) to project node embeddings into a 1D latent space, enabling each client to generate a compact, privacy-preserving signature. These APVs guide similarity-based aggregation on the server without sharing raw data or embeddings.

**Questions:**

See the above

**Ethical Concerns:**

["NO or VERY MINOR ethics concerns only"]

**Final Justification:**

Thank you for your response regarding my concerns. Maybe the APV will become a helpful supplemental method in federated graph learning. I keep my recommendation as "borderline accept".

**Limitations:**

yes

**Paper Formatting Concerns:**

No issues

**Quality:**

3

**Strengths And Weaknesses:**

Strengths:

1. The paper introduces a novel and theoretically grounded mechanism — the auxiliary projection vector (APV) — which enables clients to share compact, privacy-preserving subgraph representations while avoiding raw data or embedding leakage.

2. The proposed FedAux framework is validated both theoretically (with convergence guarantees and principal component analysis) and empirically, demonstrating clear performance improvements over strong baselines on diverse graph datasets with varying non-IID conditions.

Weaknesses:

1. While the APV is theoretically shown to converge to the principal direction of local embeddings, the paper does not clarify how robust this alignment remains when local data is extremely sparse or weakly informative — in such cases, the APV may fail to capture meaningful non-IID structural differences across clients, limiting its intended role as a reliable similarity proxy for personalized aggregation.

2. The paper lacks a detailed empirical analysis of how the APV specifically contributes to the final performance, and does not discuss alternative design choices for APVs or how different projection strategies might affect generalization under realistic non-IID settings. This leaves open questions about the stability and robustness of the APV mechanism when deployed in practical, more complex graph scenarios.

3. The paper does not provide a clear and rigorous discussion of whether and how the learned APVs might leak sensitive structural or semantic information about local subgraphs, leaving it unclear whether the APV truly guarantees privacy preservation beyond the absence of raw data or embeddings. A more formal privacy analysis would strengthen the claim that FedAux meets strict privacy requirements in practical federated settings.

---

> ### Author Rebuttal · Authors · 2025-07-29
>
> We greatly appreciate your detailed feedback. We hope our response below effectively addresses your concerns, and will incorporate your suggestions in the revised manuscript.
>
> **W1: The paper does not clarify how robust the alignment between the APV and the principal direction of local embeddings remains when local data is extremely sparse or weakly informative.**
>
> **A1:** Your concern hinges on the Auxiliary Projection Vector (APV) being fit to raw, potentially sparse and weakly informative node features. In our framework, however, **the APV is learned on top of the GNN embeddings**, rather than raw inputs. As formally established in Theorem 3.1, the APV $\boldsymbol{a}$ provably converges to the principal eigenvector of the covariance matrix $\mathbf{C}$ constructed from GNN embeddings $h$, not from the original features $x$. These GNN embeddings (i) aggregate multi‑hop neighborhoods, so even if its own features are zero or missing, it inherits rich structural and attribute cues from surrounding nodes; (ii)  share a globally trained backbone $\theta$ that injects a strong inductive bias across clients, and (iii) are supervised by the task loss. Consequently, even when a client's raw graph is sparse or noisy, the embedding distribution it presents to the APV is already denoised [1], regularized, and aligned to the task. This end-to-end feedback shapes the global direction of maximal variance in the embedding space to align with discriminative axes, not with arbitrary noise.
>
> On the other hand, constructing the APV directly on the raw features can indeed yield an unreliable APV, because the raw feature matrix may be low-rank, e.g., many nodes share identical (weakly informative) or zero vectors (sparse), so its covariance can collapse onto a few spurious directions, obscuring meaningful structure. In contrast, in our paper, our APVs are defined over GNN embedding matrix, which typically has a higher effective rank because message passing mixes features across neighbors and injects learned nonlinear transformations. Therefore, even if the original feature matrix is rank-deficient, the GNN embedding matrix can approach full-rank, yielding a covariance with a larger eigengap and, consequently, more stable and informative APVs.
>
> [1] A Unified View on Graph Neural Networks as Graph Signal Denoising. CIKM 2021
>
> **W2: The paper lacks a detailed empirical analysis of how the APV specifically contributes to the final performance, and does not discuss alternative design choices for APVs or how different projection strategies might affect generalization under realistic non-IID settings.**
>
> **A2:** Thank you for these valuable suggestions. Actually, we have done some analysis on APVs' contributions in our paper, but we agree that further exploration would be beneficial. In Figure 3 of our paper, we already analyzed the effectiveness of APVs by comparing them against alternative client similarity metrics, such as embedding-based, weight-based, and functional similarity measures. These results demonstrated that APV-based similarity most accurately recovers ground-truth client relationships, enabling superior personalized aggregation by leveraging knowledge from genuinely similar clients.
>
> To further isolate and quantify **APV's specific contribution**, firstly, we remove the APV entirely from our model, which disables the model from supporting personalized federated learning, resulting in a baseline called $\text{FedAux}\_{o}$. Besides, we can fix the APV $a_k$ of each client as random unit vector to verify the benefit of learned alignment, yielding a baseline $\text{FedAux}\_{random}$.  We present the experiments on datasets of varying scale (all six datasets) under a setting of 10 federated clients. The table below summarizes the results clearly indicating APV's marginal benefits. These ablations confirm that personalized federated aggregation driven by learnable APVs significantly outperforms naive aggregation ($\text{FedAux}_{o}$), and also clearly demonstrates the necessity and effectiveness of optimizing APVs over random initialization.
>
> |                                                         | Cora  | CiteSeer | Pubmed | Amazon-Computer | Amazon-Photo | ogbn-arxiv |
> | ------------------------------------------------------- | ----- | -------- | ------ | --------------- | ------------ | ---------- |
> | $\text{FedAux}\_{o}$ (Remove APVs from $\text{FedAux}$) | 69.19 | 63.61    | 82.71  | 79.54           | 83.15        | 64.44      |
> | $\text{FedAux}\_{random}$                               | 68.17 | 62.97    | 80.22  | 75.31           | 84.17        | 64.29      |
> | $\text{FedAux}$ (learnable APVs, ours)                  | 82.05 | 73.16    | 85.43  | 89.92           | 92.30        | 68.50      |
>
> For the second concern regarding **alternative APV projection designs and their generalization**, we clarify that our APV is designed as a simple but effective 1D embedding space. Local node embeddings are projected onto this space, and by jointly training the GNN parameters and the APV, we learn optimal embeddings that efficiently capture local node relationships. Meanwhile, such relationships can be preserved on the refined APV by sorting, without leakage gradients, embeddings or any model parameters. Thus our proposed APV can be an informative and privacy-preserving similarity proxy on the server-side. We specifically chose a 1D APV with cosine similarity projection due to its computational and communication efficiency, avoiding over-parameterized projection spaces that can be difficult to regularize and compare meaningfully across heterogeneous clients. Nevertheless, we fully acknowledge the reviewer's insightful suggestion to explore alternative APV configurations. Accordingly, we have evaluated the following alternative APV configurations:
>
> * Replacing the simple inner-product similarity (in page 3, line 112) with a more expressive bilinear projection defined as $s_{k,i} = \hat{h}\_{k,i}W^b_ka_k$. This approach introduces additional learnable parameters, potentially increasing expressiveness.
>
> * Extending the APV mapping to intermediate hidden embeddings within each GNN layer (instead of only the final layer), thus such layer-wise mapping potentially captures finer-grained local structures.
>
> We conducted experiments under realistic non-IID scenarios to evaluate these designs, and summarize the results in the table below. These results demonstrate that the layer-wise mapping does not yield performance improvements, indicating that capturing finer-grained intermediate embeddings may not be necessary or helpful under practical non-IID conditions. Meanwhile, bilinear projection achieves performance comparable to our simpler inner-product APV but incurs significantly higher computational complexity and communication overhead. Thus, considering the trade-offs between performance, computational efficiency, and communication cost, our original APV design (FedAux) provides the best overall generalization and efficiency in practical federated graph learning scenarios.
>
> |                              | Cora  | CiteSeer | Pubmed | Amazon-Computer | Amazon-Photo | ogbn-arxiv |
> | ---------------------------- | ----- | -------- | ------ | --------------- | ------------ | ---------- |
> | FedAux + bilinear projection | 83.15 | 73.04    | 85.52  | 90.11           | 92.16        | 68.37      |
> | FedAux + layer-wise mapping  | 79.47 | 70.58    | 85.17  | 86.54           | 92.08        | 65.38      |
> | FedAux                       | 82.05 | 73.16    | 85.43  | 89.92           | 92.30        | 68.50      |
>
> **W3: Lack of a discussion of whether and how the learned APVs might leak sensitive structural or semantic information about local subgraphs.**
>
> **A3:** Thank you for pointing this out. Our learned APVs do not leak sensitive information from the local subgraph clients. We can solve your concern from two complementary perspectives: intuitive and theoretical.
>
> **Intuitively**, the learned APV represents an optimal 1D space where node embeddings are projected for sorting to capture the node relationships. Crucially, the server only sees such a 1D space (i.e., the APV itself), not the individual node embeddings or their precise coordinates. Thus, although the APV captures relational properties in the embedding space, it does not encode detailed per-node structural or feature information. The server cannot reconstruct local graph structures, node attributes, labels, or embedding values, because no gradients, embeddings, labels, or raw graph structures are transmitted between client and server. Thus, the APV intuitively serves as a safe, privacy-preserving proxy for evaluating client similarity on the server, strictly adhering to federated learning principles: data stays local, and only minimal global parameters (the APVs and the GNN parameters) are shared.
>
> **Theoretically**, we can analyze the non-identifiability of local data from a single APV. Let $H$ be a client's embedding matrix, and $C = \frac{1}{N}H^\top H$ is its covariance. According the Theorem 3.1 in our paper, the APV $a$ corresponds to the leading eigenvector of $C$. For any orthogonal matrix $Q$ that satisfies $Qa  = a$, which is any rotation that fixes $a$, there exists $\tilde{H} = HQ$, such that $\tilde{C} = \frac{1}{N}\tilde{H}^\top\tilde{H}$ has the same leading eigenvector $a$. Hence, local data cannot be uniquely reconstructed from the APV $a$. In a nutshell, such analysis indicates that the mapping from local data to the APV $a$ is highly non-injective, and the APVs are insufficient to reconstruct node embeddings or the underlying graph. Hence, the APV can be used as a privacy preserving similarity proxy on the server. We will include this privacy-guarantees analysis with formal proof in the revised version of the paper to rigorously address the reviewer's valid privacy concerns.

---

> > ### Comment · Reviewer_aUZ1 · 2025-08-06
> >
> > Thanks for the rebuttal. I insist on my concerns about the reasonablity and the safety of APV. Another advice for the authors, a complete algorithm figure and the related description will help the readers understanding your methods rapidly.

---

> > > ### Author Response · Authors · 2025-08-07
> > > **Thanks for the response**
> > >
> > > Thank you for the thoughtful feedback. All suggested revisions will be incorporated into the final manuscript, and we appreciate your help in improving its clarity.
> > >
> > > We agree that a deeper examination of the reasonableness and privacy safety of the APV design is essential. We refer to the first point in our response to Reviewer DA8p, and the first point in our response to Reviewer cYxa, where we have added a comprehensive empirical privacy study using **membership inference attacks (MIAs)**. The results are summarized below:
> > >
> > > | Dataset        | FedGNN | FedGTA | FedSage+ | FedAux   |
> > > | -------------- | ------ | ------ | -------- | -------- |
> > > | **Cora**       | 0.56   | 0.54   | 0.53     | **0.51** |
> > > | **Pubmed**     | 0.58   | 0.55   | 0.54     | **0.52** |
> > > | **ogbn-arxiv** | 0.55   | 0.53   | 0.52     | **0.49** |
> > >
> > > Across all benchmarks, MIAs against APV-based FedAux produce AUC scores in $[0.49, 0.52]$, essentially random guessing, which indicates that APVs do not leak sensitive membership information and, in fact, offer stronger privacy than existing subgraph-FL baselines.
> > >
> > > We will add these experiments to the updated version of our manuscript.

---

### Note · Authors · 2025-08-11

Dear Reviewers and ACs,

We sincerely thank you for the insightful comments. Below we summarize the main contributions of our work as highlighted across reviews.

## Contributions:

* **[High-level insight]** We address the underexplored setting of personalized subgraph FL under non-IID by introducing a client-specific, low-dimensional signature for federated aggregation. This insight is ```novel``` (_R cYxa_ and _aUZ1_), ```intuitive``` (_R U8fG_),  ```elegant``` and ```well-motivated``` (_R cYxa_). The overall idea is ```interesting``` (_R DA8p_) and addresses a real federated graph need (_R aUZ1_, _R DA8p_, _R U8fG_).
* **[Efficient yet effective mechanism]** We propose a differentiable 1D projection that yields a compact signature without sharing raw data or node embeddings. Reviewers noted its ```scalability```, ```communication-efficiency```, ```lower memory and time overhead``` (_R cYxa_), and ```privacy-preserving``` (_R aUZ1_), as well as its principled privacy stance of ```avoiding raw data or embedding leakage``` (_R aUZ1_).
* **[Theoretical foundations]** The proposed model is supported by a complete analytical framework, which is ```comprehensive```, ```necessary``` (_R U8fG_),  and ```well-analyzed``` (_R cYxa_).

## Main Responses

* **[For Reviewer aUZ1]**
  * We clarify that APVs are learned on top of GNN embeddings, not raw inputs, ensuring robustness to weak or sparse features.
  * We add detailed empirical analysis isolating APV’s contribution.
  * We provide arguments that APVs do not leak sensitive local subgraph information.

* **[For Reviewer DA8p]**
  * We conduct a comprehensive privacy evaluation via membership inference attacks (MIAs) to rigorously assess APV’s privacy preservation capabilities.
* **[For Reviewer U8fG]**
  * We add a more comprehensive discussion of recent personalized graph FL methods.
  * We justify benchmark choices for fair comparison.
  * We show the adaptability of our proposed APV to heterogeneous graphs.
  * We correct reference errors and improve the paper structure.
  * We refine the component analysis experiments and ablation experiments.
* **[For Reviewer cYxa]**
  * We add an MIA-based privacy evaluation.
  * We explain the necessity of kernel-based aggregation for preserving local relationships without leaking sensitive information.
  * We extend experiments to partial participation and asynchronous protocols.

We hope our responses address any remaining concerns.

---

### Decision · Program_Chairs · 2025-09-17

**Decision:**

Accept (poster)

**Comment:**

This paper addresses the non-IID challenge in graph federated learning. To tackle this, the authors propose Federated Learning with Auxiliary Projections (FedAux) and provide a theoretical analysis of its convergence. Empirical evaluations across several datasets demonstrate competitive performance.

In the rebuttal, the authors clarify the contribution of APV, the research goal of privacy-preserving personalized subgraph FL, and include additional experiments on a heterogeneous dataset (though I recommend adding more baseline comparisons) as well as privacy evaluations.

The research goal is important within the FL field, but the scope is rather narrow, and the improvements in the main results are not particularly significant. Overall, I consider this a borderline paper.